# Molecular architecture of the yeast Mediator complex

**Philip J Robinson[1†], Michael J Trnka[2†], Riccardo Pellarin[3,4], Charles H Greenberg[3], David A Bushnell[1], Ralph Davis[1], Alma L Burlingame[2], Andrej Sali[3], Roger D Kornberg[1]***

[1]Department of Structural Biology, Stanford University School of Medicine, Stanford, United States; [2]Department of Pharmaceutical Chemistry, University of California, San Francisco, San Francisco, United States; [3]Department of Bioengineering and Therapeutic Sciences, Department of Pharmaceutical Chemistry, California Institute for Quantitative Biosciences, University of California, San Francisco, San Francisco, United States; [4]Structural Bioinformatics Unit, Institut Pasteur, CNRS UMR 3528, Paris, France

**Abstract** The 21-subunit Mediator complex transduces regulatory information from enhancers to promoters, and performs an essential role in the initiation of transcription in all eukaryotes. Structural information on two-thirds of the complex has been limited to coarse subunit mapping onto 2-D images from electron micrographs. We have performed chemical cross-linking and mass spectrometry, and combined the results with information from X-ray crystallography, homology modeling, and cryo-electron microscopy by an integrative modeling approach to determine a 3-D model of the entire Mediator complex. The approach is validated by the use of X-ray crystal structures as internal controls and by consistency with previous results from electron microscopy and yeast two-hybrid screens. The model shows the locations and orientations of all Mediator subunits, as well as subunit interfaces and some secondary structural elements. Segments of 20–40 amino acid residues are placed with an average precision of 20 Å. The model reveals roles of individual subunits in the organization of the complex.

*For correspondence:
kornberg@stanford.edu

†These authors contributed equally to this work

**Competing interests:** The authors declare that no competing interests exist.

## Introduction

Mediator is a complex of at least 21 subunits, with a mass of greater than 1 million Daltons, conserved from yeast to humans (reviewed in *Kornberg, 2005*; *Conaway and Conaway, 2011*; *Ansari and Morse, 2013*). Genetic studies have shown that Mediator is essential for RNA polymerase II (pol II) transcription and required for positive and negative regulation of transcription. Biochemical studies have confirmed the importance of Mediator for both basal and regulated transcription, and have demonstrated interactions of Mediator with pol II, with general transcription factors, and with transcriptional activator proteins. Through these interactions, Mediator is thought to promote assembly of the entire machinery for pol II transcription.

Electron microscopy (EM) has shown a division of Mediator in three modules, termed Head, Middle, and Tail (*Asturias et al., 1999*). EM studies further showed that Mediator structure is conserved from yeast to humans, that Mediator is compact when free in solution, and that a major structural rearrangement occurs upon interaction with pol II, which is surrounded by the Head and Middle modules in the so-called 'holoenzyme' (*Asturias et al., 1999*; *Dotson et al., 2000*). Details of subunit interaction networks within Mediator modules have come from co-expression studies (*Koh et al., 1998*; *Lee et al., 1998*; *Kang et al., 2001*; *Koschubs et al., 2010*) and from Mediator-specific (*Guglielmi et al., 2004*) and genome-wide (*Uetz et al., 2000*; *Ito et al., 2001*) two-hybrid

**eLife digest** Inside a cell, proteins are made from instructions encoded by DNA. To produce a particular protein, a section of DNA within a gene is copied into a molecule of messenger ribonucleic acid (or mRNA). This process is called transcription and is carried out by an enzyme known as RNA polymerase.

Transcription begins in a region of DNA called a promoter, which is found at the start of the gene. RNA polymerase is brought to the DNA by many proteins, including the so-called Mediator complex. Mediator receives signals from within the cell and from the environment, processes the information, and instructs RNA polymerase whether to transcribe the gene or not. Mediator performs this important role in all organisms from yeast to humans, but it is not clear how it works. A crucial step towards the solution of this problem is to understand the three-dimensional structure of the complex.

Previous research using a technique called 'electron microscopy' showed that Mediator is composed of three modules, referred to as Head, Middle and Tail. The images from electron microscopy were not sufficiently detailed to reveal the organization of the proteins within these modules. An open-source Integrative Modeling Platform (IMP for short) was recently developed to arrive at structural models of large protein complexes from a combination of experimental data and computer models. Now, Robinson, Trnka, Pellarin et al. have used this platform to study the Mediator complex.

First, Robinson, Trnka, Pellarin et al. collected experimental data on the structure of the Mediator complex using two approaches called 'chemical cross-linking' and 'mass spectrometry'. This data was combined with biochemical and structural information from previous studies to generate a three-dimensional model of the structure of the entire Mediator using IMP. The model is detailed enough to show the location and orientation of all the proteins in the complex. For example, a protein called Med17 connects the Head and Middle modules, while another subunit—known as Med14—spans the entire complex and makes extensive contacts with other proteins in all three modules.

approaches. The currently accepted subunit assignments are Med 6-8-11-17(Srb4)-18(Srb5)-20(Srb2)-22(Srb6) in the Head module, Med 1-4-7-9(Cse2)-10(Nut2)-19(Rox3)-21(Sbr7)-31(Soh1) in the Middle module, and Med 2-3(Pgd1)-5(Nut1)-14(Rgr1)-15(Gal11)-16(Sin4) in the Tail module (old naming convention in parentheses).

X-ray crystal structures of Head modules from *Saccharomyces cerevisiae* and *Schizosaccharomyces pombe* have been determined, with (*Robinson et al., 2012*) and without (*Imasaki et al., 2011*; *Lariviere et al., 2012*) the carboxy-terminal domain (CTD) of the largest pol II subunit, Rpb1, bound to a conserved site on the surface. CTD interaction with Head and Middle modules plays an important role in the association of Mediator with pol II and the pre-initiation complex. Beyond the X-ray crystallography of the Head module, structural information on Mediator is limited. Only small portions of the Middle (*Baumli et al., 2005*; *Koschubs et al., 2009*) and Tail modules (*Bontems et al., 2011*; *Eletsky et al., 2011*; *Vojnic et al., 2011*) have been solved at near atomic resolution. EM at low resolution with subunit-labeling has given an approximate idea of the spatial organization of the modules (*Tsai et al., 2014*; *Wang et al., 2014*), and an architectural model for a subcomplex of the Middle module (lacking Med 1 and 19) has been proposed on the basis of cross-linking data and homology modeling (*Lariviere et al., 2013*). Recently, reconstituted yeast Head and Middle Mediator modules were visualized by EM in a complex with pol II and a subset of general transcription factors (*Plaschka et al., 2015*).

Mediator is representative of a large number of macromolecular complexes that are refractory to classical high-resolution structural approaches, due to low native abundance and to conformational and compositional heterogeneity. For the study of such systems, data from chemical cross-linking and mass spectrometry can provide multi-protein interaction maps at residue-level resolution (*Murakami et al., 2013*; *Erzberger et al., 2014*; *Shi et al., 2014*). The approach is particularly effective when combined with other structural information, such as electron density maps from EM (*Murakami et al., 2013*) and subunit structures from X-ray crystallography (*Robinson et al., 2012*). To construct structural models of macromolecular assemblies by combining such diverse types of information, the

open-source Integrative Modeling Platform (IMP) program was developed (http://integrativemodeling.org) (*Russel et al., 2012*). IMP translates the data into spatial restraints, computes an ensemble of models by satisfying these restraints as well as possible, and finally assesses the ensemble against the data used or not used in its construction (Schneidman-Duhovny et al., 2014). IMP has been successfully applied to a number of protein complexes (*Lasker et al., 2012*; *Erzberger et al., 2014*; *Shi et al., 2014*).

Here we determined the molecular architecture of Mediator by integrative structure determination, based on chemical cross-links, X-ray crystal structures, homology models, and a cryo-EM electron density map, with the use of IMP. Possible configurations of the Mediator subunits were exhaustively sampled to identify those that best satisfied the experimental restraints. The resulting model was validated by re-computing it with random subsets of chemical cross-links (i.e., 'jackknifing') (*Brunger et al., 1993*) as well as by comparison with known protein structures and protein interaction networks (*Guglielmi et al., 2004*). The Mediator model will serve as a basis for studies of Mediator interaction with pol II, general transcription factors, gene activators, and gene repressors.

## Results

### Isolation and cross-linking of native Mediator

Mediator was isolated from yeast as originally described (*Kim et al., 1994*) in the form of a complex with pol II (holoenzyme). The complex is not only a functionally relevant form of Mediator, but is also more soluble than free Mediator under the conditions used for lysine directed cross-linking. We employed a double affinity-tagging strategy (*Figure 1—figure supplement 1*) and obtained 20-fold more stoichiometric holoenzyme than from previous purification procedures. The native complex was stable, persisting throughout purification, whereas a complex formed from separately isolated Mediator and pol II was disrupted by the same purification procedures.

Cross-linking and mass spectrometry were performed on the native holoenzyme, with a mixture of D12-labelled and unlabeled BS3 cross-linking reagents, or with unlabeled BS3 and enrichment of cross-linked trypsin fragments by gel filtration. The BS3 reagent bridges free-amine moieties at surface lysines and N-termini with $C\alpha$ spacing within approximately 25 Å. Cross-links were assigned with the Protein Prospector platform and a refined scoring function (*Trnka et al., 2014*). The 320,767 initial product ion spectra were searched against a database containing the sequences of the yeast transcriptional machinery as well as 520 randomized decoy sequences, identifying 20,388 potential cross-linked spectra (*Figure 1A*). Classification of these matches and the application of quality filters resulted in the identification of 3196 holoenzyme cross-linked spectra at a false discovery rate (FDR, fraction of decoy hits above score cutoff) of 0.7% (*Figure 1A*). These spectra defined 402 unique pairs of linked amino acid residues ('cross-links') with a FDR of 4.1% (*Figure 1—source data 1*, *Figure 1—figure supplement 2*, *Supplementary file 1*). The majority (260 cross-linked fragments) were between Mediator proteins, 124 were between pol II subunits, and 18 were between Mediator and pol II.

### Validation of holoenzyme cross-link dataset

The pol II component of the holoenzyme provided an internal control, enabling validation of the cross-link dataset by two approaches. First, the X-ray crystal structure of pol II and also that of the Mediator Head module could be used to assess the validity of cross-links, as 130 of the 402 high confidence cross-links were between residues in these structures. Only five of these 130 cross-links occurred between residues further apart than the 35 Å threshold (*Figure 1B*). This number of violations was therefore consistent with the FDR of 4.1%, estimated from the number of decoy database hits. Second, we constructed an integrative model of pol II based on the cross-link dataset, a cryo-EM electron density map (EMD-1883, 20.9 Å resolution using FSC 0.5), and X-ray crystal structures of the pol II subunits, using IMP. The X-ray crystal structures of the subunits were represented as independent rigid bodies, and the regions not observed in the structure were represented by flexible strings of beads. A low resolution EM map was used for comparison with the EM data available for Mediator (see below). Starting from scrambled configurations, the positions and orientations of the pol II subunits were optimized with a Monte Carlo procedure (which applies a large number of random movements guided by the input data), resulting in configurations that satisfied the input data well. This ensemble of good-scoring solutions recapitulated the known architecture of pol II (*Figure 1C*).

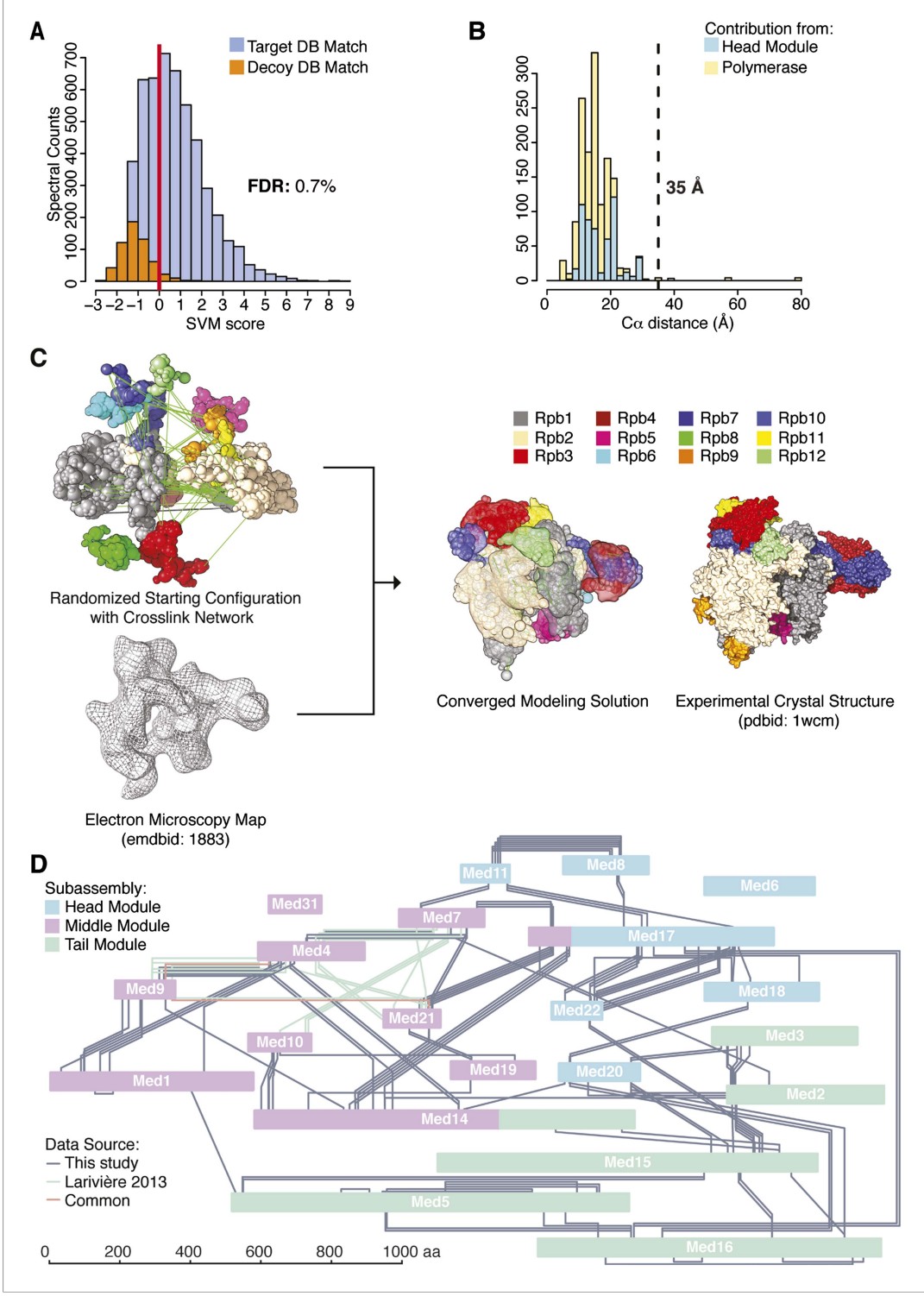

**Figure 1**. Holoenzyme cross-linking and modeling results and methodology. (**A**) Cross-links were identified by searching MS2 product ion spectra against concatenated target and decoy databases containing 52 *Saccharomyces cerevisiae* transcription proteins +520 sequence randomized versions, respectively. Confidence in the spectral assignment is represented by an support vector machine (SVM) classifier (*Trnka et al., 2014*). The distribution of the target and decoy spectral matches in relation to the acceptance threshold (red line) is shown with an overall spectral false discovery rate (FDR) of 0.7%. (**B**) Mapping identified cross-linked spectra onto regions of known structure such as the Mediator Head module and RNA polymerase II yields distance distributions that reflect the accuracy of

*Figure 1. continued on next page*

*Figure 1. Continued*

cross-link identification. The Cα violation distance of 35 Å is indicated with a dashed line. (**C**) Demonstration of RNA pol II structural recapitulation from a random starting configuration (top panel) of individual pol II subunits represented as rigid bead models and restrained by cross-links (green links) and electron microscopy (EM) density map (mesh). A representation of the converged modeling solution is presented alongside the 12-subunit pol II X-ray structure (PDB: 1WCM) for reference. (**D**) Schematic representation of the Mediator complex cross-linking network. Mediator subunits are colored according to their location within the Head, Middle and Tail modules. All Inter-subunit cross-links and selected intra-subunit cross-links from the current study as well from (*Lariviere et al., 2013*) are represented, colored according to their origin.

The following source data and figure supplements are available for figure 1:

**Source data 1**. Categorization of cross-links with respect to module and data source.

**Figure supplement 1**. Purification of Native *S. cerevisiae* Holoenzyme complex.

**Figure supplement 2**. Annotated product ion spectra for selected cross-linked peptides.

The average root-mean-square deviation (RMSD) of the solutions with respect to the crystal structure of pol II was 14 Å. All subunits were correctly placed, except for Rpb8 and Rpb9, whose locations were undetermined, due to a lack of cross-link restraints.

## The N-Termini of Med14 and Med17 are components of the Mediator Middle module

We augmented our Mediator cross-link dataset with 38 Mediator Middle module cross-links from a dataset collected on a recombinant preparation of *S. cerevisiae* Middle module containing 6 subunits (*Lariviere et al., 2013*). The 38 cross-links were selected according to our minimal length requirement of four amino acid residues per cross-linked peptide; three cross-links were common to both datasets. This modest overlap is likely due to the use of a different chemical cross-linker, as well as to the presence of additional subunits in our full native Middle module. The complete dataset of 294 inter-subunit cross-links defined an interaction map of 19 of the 21 Mediator proteins (*Figure 1D*). The two subunits for which there were no inter-subunit cross-links have been crystallized in complexes with other Mediator subunits: Med6 as part of the Head module (*Robinson et al., 2012*), and Med31 in a complex with the N-terminal region of Med7 (*Koschubs et al., 2009*). Both proteins were present in our native holoenzyme preparations, as shown by SDS PAGE and mass spectrometry (*Figure 1—figure supplement 1*). Two subunits, Med17 (Head module) and Med14 (Tail module) showed patterns of cross-links split between two or more different Mediator modules. The majority of Med17 (residues 123–687) formed an extensive cross-link network with other subunits of the Head module, as expected from the X-ray crystal structure (*Robinson et al., 2012*). The N-terminal region of Med17 (1–122), not observed in the Head module crystal structure, formed many cross-links with Med7 and Med21, subunits of the Middle module (*Figure 1D*). Similarly, Med14, a presumptive Tail module subunit, formed cross-links to other Tail module proteins (Med2 and 15), but almost exclusively in its C-terminal region (residues 712–1082), while the N-terminal region and the majority of the protein (residues 1–711) cross-linked instead to Middle module subunits (Med4, Med9, Med10, and Med21). The N-terminal regions of Med17 and Med14 not only extended into the Middle module but also formed many cross-links with each other. For these reasons, we redefine the Mediator modules to include the N-terminal domains of Med17 (residues 1–122) and Med14 (residues 1–711) in the Middle module rather than the Head and Tail modules (*Figure 1D*, Med14 and Med17 colored by module localization).

## Integrative modeling of Mediator complex

Although cross-linking was performed on the holoenzyme, about 90% of the cross-links in our dataset were within Mediator modules (intra-modular) or within pol II. Due to the paucity of cross-links between Mediator and pol II, and because a cryo-EM map at sufficient resolution for molecular

modeling was only available for Mediator without pol II, modeling calculations were performed for Mediator alone. The four-step modeling ('integrative structure determination') procedure (*Figure 2—figure supplement 1*) entailed: (1) gathering data; (2) representing and translating the data into spatial restraints; (3) sampling the conformational space and identifying good scoring solutions; and (4) analyzing and assessing the ensemble of solutions. All IMP input files, scripts, and output models for this study are available at http://salilab.org/mediator/.

In step 1, the data comprised our holoenzyme cross-link dataset, X-ray crystal structures of some subunits and subunit regions, homology models for some subunits, and the highest resolution cryo-EM map of free Mediator available (*Tsai et al., 2014*).

In step 2, we constructed a set of 21 Mediator subunit model representations (*Figure 2—figure supplement 2*). Atomic models from X-ray crystallography or homology models were represented by rigid bodies (*Figure 2—figure supplement 2*, blue subunit shading), and unmodeled subunit regions were represented as flexible strings of beads (*Figure 2—figure supplement 2*, yellow subunit shading). The entire Head module was represented as a rigid body (based on the X-ray structure PDB 4GWP), with the exception of the unmodeled N-terminus of Med17 (1–181), the last 103 residues of Med6, and other missing regions, all of which were represented as flexible strings of beads. Manual docking and the mapping of Med 8, 18, and 20 onto 2-D EM projections previously supported a single docking solution for the Head module in the EM density (*Tsai et al., 2014*). We performed an exhaustive docking calculation, comparing all possible Head module locations with a cross-correlation scoring function (Materials and methods). We found a docking solution that is consistent with the solution proposed previously, with a normalized cross-correlation coefficient approximately 40% better than the next best solution. We modeled the Mediator complex with the Head module fixed in a single position corresponding to the highest scoring docking solution. Other regions defined by atomic models were Med7C-Med21 and Med7N-Med31, represented by rigid bodies based on their X-ray structures (*Baumli et al., 2005*; *Koschubs et al., 2009*), and Med4-Med9, based on a homology model (*Lariviere et al., 2013*). In addition, we created a homology model of the first 540 residues of Med16, also in a rigid body representation, based on a seven-bladed β-propeller fold (proposed and justified below).

The EM map was divided into segments to reflect the modular organization of the complex. Since the Head module was docked and fixed in a portion of the density map, the remaining density was segmented into regions corresponding to the Middle and Tail modules on the basis of previous approximate localization of Middle and Tail module subunits within two distinct regions of the EM map (*Tsai et al., 2014*; *Wang et al., 2014*). During modeling, each Middle or Tail subunit was restrained to the corresponding Middle or Tail EM density. EM spatial restraints were based on a Gaussian decomposition of the electron density (Materials and methods).

The cross-linking data was encoded in a Bayesian scoring function (*Erzberger et al., 2014*; *Shi et al., 2014*). The scoring function allowed the independent optimization of the relative weights of arbitrary subsets of the cross-linking data. We split the cross-link dataset into two subsets composed of inter-module (8% of the total) and intra-module (92% of the total) cross-links (*Figure 1—source data 1*).

In step 3, we conducted a large number of independent modeling runs. The positions and orientations of rigid bodies and flexible strings of beads were repeatedly perturbed in an effort to satisfy the scoring function consisting of the excluded volume, sequence connectivity, EM, and cross-linking restraints. A total of 165,523 Mediator model configurations were produced. The 500 best-scoring models (solutions) were grouped on the basis of RMSD into four clusters (C1-4, *Figure 2—figure supplement 3*), the minimum number required to fully represent the main structural differences between the best scoring models. To confirm that we had sampled conformational space sufficiently to reach model convergence, we compared two independent halves of the solutions to each other and to the entire set, showing they were all similar to one another (*Figure 2—figure supplement 4*).

The solutions satisfied all excluded volume, sequence connectivity, and EM restraints. Intra-module cross-links were satisfied to a high degree (approximately 95%), and inter-modular cross-links to a much lesser degree (about 10%) (*Figure 2—figure supplement 3*). In fact, the Bayesian scoring function automatically assigned a low scoring weight to the inter-modular cross-links, pointing to an inconsistency of the inter-modular cross-links in the holoenzyme dataset with the EM density map of the free Mediator. The satisfaction of intra-modular cross-links suggests that the arrangement of

Mediator subunits within modules of the holoenzyme is similar to that in the free Mediator, whereas the inconsistency of the inter-modular cross-links with the EM map suggests that motions of the modules relative to one another occur upon association with pol II in the holoenzyme. Indeed, EM studies have indicated large motions about hinges between the modules, opening the compact structure of free Mediator for wrapping around pol II in the holoenzyme (*Asturias et al., 1999*; *Davis et al., 2002*). Based on these observations, the inter-modular cross-links were not considered in our analysis of the free Mediator.

## A single, best-scoring subunit architecture

The four clusters of best-scoring solutions differed in two respects. Cluster 3, which represented a minor subset of best-scoring models (~8%), differed from the other clusters by an inversion of orientation of the Middle module. This cluster was disregarded because the inverted orientation was inconsistent with prior EM localization experiments (*Tsai et al., 2014*) and because of significantly worse intra-module cross-link violation statistics (*Figure 2—figure supplement 3D–E*). The remaining three clusters were virtually identical in the locations of all subunits except for Med5, Med15, and Med16 in the Tail module (*Figure 2—figure supplement 3C,G*). Only Cluster 1 showed an arrangement of these three subunits consistent with previous EM localization (*Figure 2—figure supplement 3H*). Cluster 1 also showed the best cross-link satisfaction statistics and had the best average score. Cluster 1 was therefore taken to represent the true Mediator subunit architecture.

The overall precision of our integrative model, computed as the average RMSD of the solutions in Cluster 1 with respect to the cluster center, is 19 Å. The precision of the Middle and Tail modules is 24 and 33 Å, respectively. These values represent the average fluctuations of the individual residues or beads in 3-D space across the ensemble of solutions comprising the highest scoring cluster. Precision defined in this way is not directly comparable to the resolution of an EM map, which refers to the uncertainty of the overall electron density, without regard to the placement of components within the density. Although the precision of the Tail module is lower than that of the Middle or Head modules, it is sufficient to position the Tail subunits within the Tail EM density as features with high cross-link density, and consequently high local precision values (such as the Middle-Tail junction), provide strong spatial restraints. The overall precision of the Mediator model sufficed to determine the positions and orientations of all Mediator subunits, and 20–40 residue beads could be localized with precisions of 10–50 Å (*Figure 2—figure supplement 5*), depending on the density of cross-linking restraints. We represented the cluster of solutions by a localization density (*Figure 2*), defined as the probability of observing a particular subunit at a specific point in space in the cluster of solutions. Because the model is three-dimensional, it depicts the contact surfaces between subunits, with residue-level resolution at many points where cross-links are formed between them. This contact information is especially rich and reliable in regions where multiple cross-links are formed and where contacts occur in multiple models in the cluster, as depicted by the darkest boxes in the domain interaction map (*Figure 3*). Some cross-links anticipated from the domain interaction map were not observed, most likely because cross-linking depends on the occurrence of unprotonated, appropriately oriented, solvent–accessible pairs of lysine residues, and on the formation of tryptic peptides with physicochemical properties amenable to liquid chromatography and mass spectrometry.

## Validation of the Mediator Model

The Mediator model (*Figure 2A*) fits the EM map used as a modeling restraint (*Figure 2B* and *Figure 7—figure supplement 1*), satisfied most of the cross-link restraints (*Figure 2—figure supplement 3*), and was consistent with almost all data from previous subunit interaction and subunit localization studies (*Figure 2C–E*). The model explains subunit interactions deduced from co-expression trials (*Kang et al., 2001*; *Beve et al., 2005*; *Koschubs et al., 2010*), pulldowns/immunoprecipitation (*Kang et al., 2001*; *Zhang et al., 2004*), and yeast two-hybrid analyses (*Uetz et al., 2000*; *Ito et al., 2001*; *Guglielmi et al., 2004*) (*Figure 2D*, green lines). There were only three discrepancies (*Figure 2D*, red lines), corresponding to the Med3-Med21, Med1-Med7, and Med1-Med5 interactions. The Med3-Med21 interaction, from two-hybrid data, is clearly spurious, because the proteins are found in different Mediator modules (Head and Tail) and our model, consistent with an EM localization study (*Tsai et al., 2014*), places the two subunits more than 100 Å apart. A second subunit interaction that is unsupported by our model, between the C-terminal domain of Med1 and

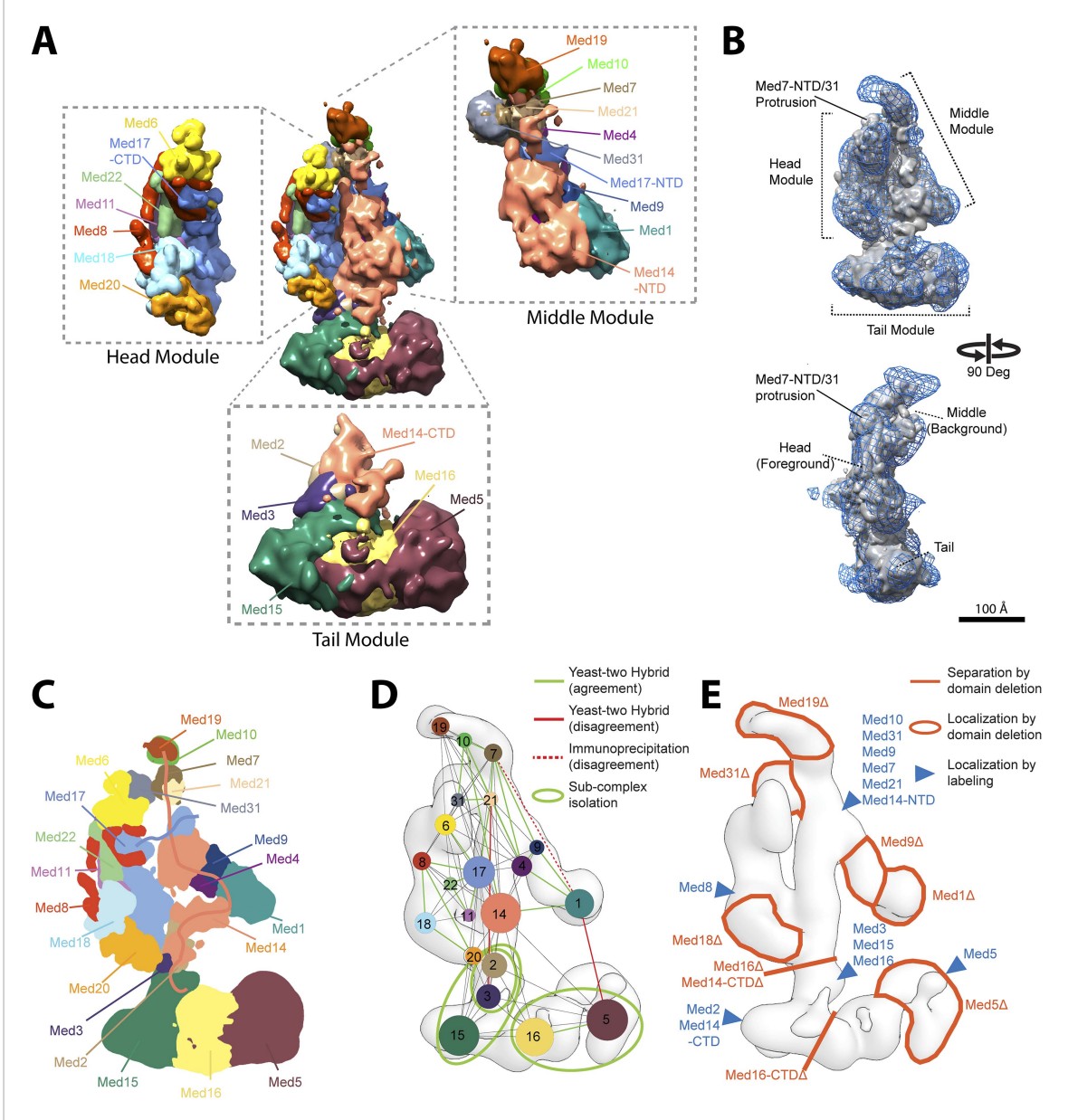

**Figure 2**. Mediator complex architecture. (**A**) Mediator subunit localization density map colored by individual subunit. (**B**) Mediator localization density map (solid grey) calculated from the highest scoring model cluster and shown at a threshold level (T = 0.1) that most closely matches the volume of the EM density map used as 3D spatial restraint during modeling (EMD-2634, T = 0.35: Blue mesh). The position of the three Mediator modules (Head, Middle, and Tail) is indicated. (**C**) Schematics of the architecture of the Mediator complex obtained from the localization density analysis. Med14 and Med17-NTD are also schematically represented by splines. (**D**) Protein–protein interactions derived from published Yeast-two hybrid, immunoprecipitation and sub-complex isolation data (*Uetz et al., 2000*; *Ito et al., 2001*; *Kang et al., 2001*; *Guglielmi et al., 2004*; *Zhang et al., 2004*; *Beve et al., 2005*; *Koschubs et al., 2010*). The data is represented by a graph superposed on the EM density map of the complex (*Tsai et al., 2014*). Nodes are Mediator subunits, the edges are the observed proteomic interactions, and the green circles are isolated sub-complexes. Green and red edges are interactions that are in agreement and in disagreement with the Mediator model, respectively. (**E**) Localization by labeling (blue triangle) and domain deletion (areas encircled by light red closed lines) mapped on the EM density map (*Tsai et al., 2014*; *Wang et al., 2014*). Straight light-red lines represent domain deletion assays that split the complex.

The following figure supplements are available for figure 2:

**Figure supplement 1**. Schematic of integrative structure determination highlighting the individual data inputs and the four stages in our approach.

*Figure 2. continued on next page*

*Figure 2. Continued*

**Figure supplement 2**. Input Model Representations.

**Figure supplement 3**. Cluster analysis of the solution ensemble.

**Figure supplement 4**. Exhaustiveness of sampling and robustness of cross-link data.

**Figure supplement 5**. Representation of subunit position precision.

Med5, also comes from two-hybrid data (*Guglielmi et al., 2004*). Although it also relates proteins located in different modules (Middle and Tail), the two proteins are less than 30 Å apart in our model. The holoenzyme cross-link dataset includes one cross-link of Med5 (Tail) to Med1 (Middle) and two cross-links of Med5 (Tail) to Med15 (Tail). Because the Med1-Med5 cross-links are inter-modular, they

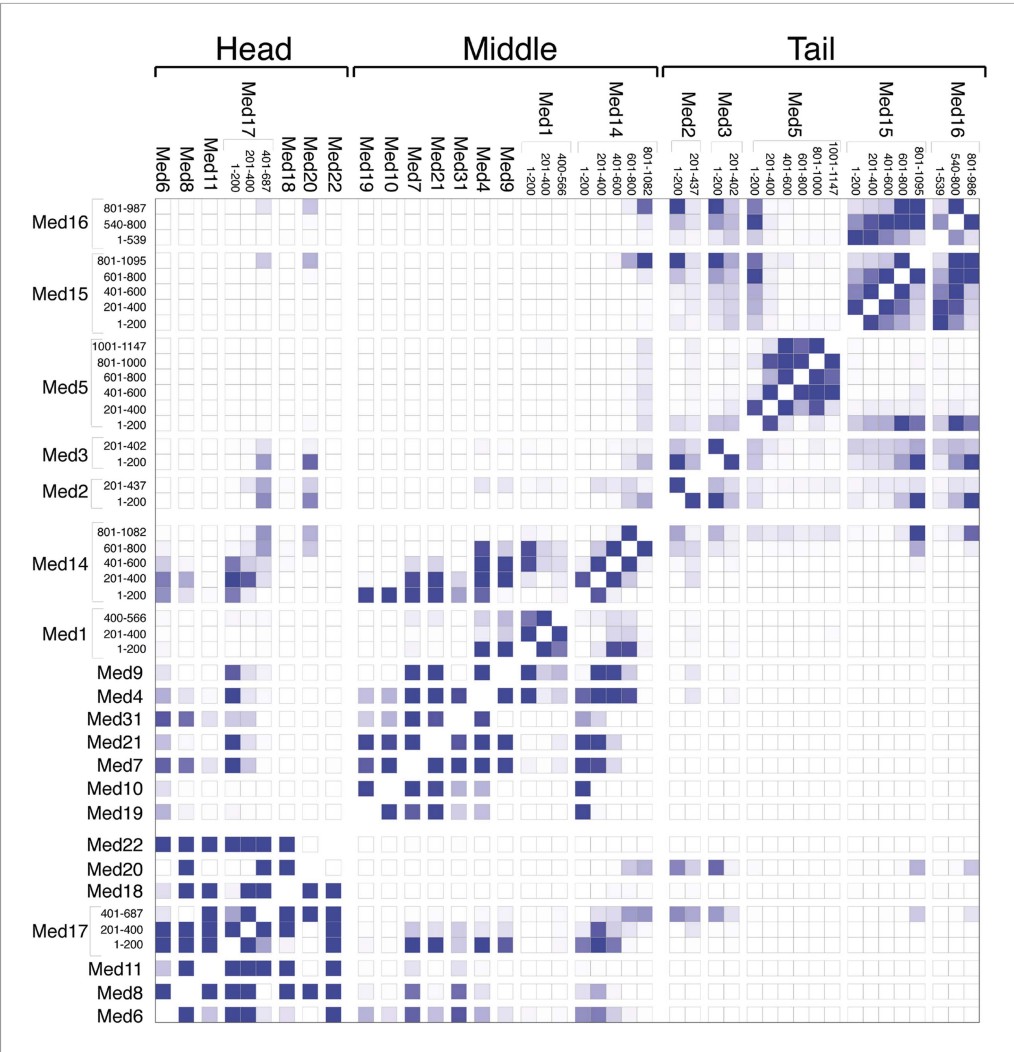

**Figure 3**. Mediator subunit domain interaction matrix. Average domain–domain contact map calculated for the cluster of best-scoring solutions. Long sequences are divided into domains of 200 residues. The intensity of the color for each box is proportional to the fraction of models for which the contact between corresponding domains is formed. Two domains are in contact when the surface of the beads of one domain is within 10 Å from the surface of any of the beads of the other domain.

were assigned lower weights by our scoring function and were not satisfied in the final model. It is possible that the interactions occur in alternative conformational states, where Med1 is in close proximity to Med5. A third subunit interaction that is unsupported by our model, between Med1-Med7, comes from immunoprecipitation data (*Kang et al., 2001*), and cannot be ruled out, because the N-terminus of Med1 is in the vicinity of the 16 C-terminal residues of Med7, which are unmodeled in the crystal structure and not well localized in our study (*Figure 2—figure supplement 5*). A lack of lysine residues in the 16 C-terminal residues of Med7 might explain why this putative interaction was not detected by cross-linking.

Our Mediator model was also validated by comparison with results of previous EM studies that mapped subunit terminal tags or subunit deletions onto 2-D class averages (*Tsai et al., 2014*; *Wang et al., 2014*). Our detailed 3-D model was consistent with all 2-D mapping results (compare *Figure 2C,E*). It also confirms a proposal that a Med7C-Med21-Med4-Med9 tetramer serves as a structural backbone within the central portion of the Middle module (*Lariviere et al., 2013*) (*Figure 4*).

## Middle module architecture

The proposed structure of the Med7C-Med21-Med4-Med9 tetramer was based on a set of 40 cross-links, an X-ray crystal structure of Med7C-Med21, and Med4-Med9 modeled as a structural homolog of Med7C-Med21 (*Lariviere et al., 2013*). Med7C and Med21 form a four-helix bundle, proposed to stack end-to-end on a four-helix bundle of Med4 and Med9, in a manner analogous to the packing observed in Med7C-21 crystals. In our modeling, we did not impose end-to-end stacking, but rather represented the Med7C-21 crystal structure and the Med4-Med9 homology model as two distinct rigid bodies. In our final Mediator model, the ends of the two four-helix bundles are in close proximity (*Figure 4B,C*), although without enforcing end-to-end helical stacking. Satisfaction of the EM restraints in other regions of the Middle module appears to require a distortion of the putative end-to-end interaction (*Figure 4C*), and many solutions displayed a twist of one four-helix bundle with respect to the other, reflecting the asymmetrical nature of the Middle module EM density.

The Med7C-Med21-Med4-Med9 tetramer provides a structural scaffold for the association of all remaining Middle module subunits (*Figure 4B,C*). The Middle module can be subdivided into two halves: Med7C-Med21, with interacting proteins Med10, Med19, and Med31; and Med4-Med9, with interacting protein Med1. The only two proteins that interact extensively with both halves of the Middle module are Med14 and Med17, which perform special architectural roles in the Mediator complex, discussed in more detail below. A Med7N-Med31 heterodimer, connected to the helical Med7C domain by an unstructured 27 amino acid residue linker, is localized to a distinct protrusion within the central portion of the EM map. The flexible C-terminal domain of Med4 is localized to the same region. The Med4-9 half of the Middle module is defined by a close association of the two large subunits Med1 and Med14 with opposite faces of the helical backbone. The N-terminus of Med1 forms an extensive interaction surface with Med4-9 (*Figures 3, 4*), with eight cross-links between residues in the first 418 residues of Med1 and residues of Med4-9 located on one face of the N-terminal four-helix bundle (*Figures 1D, 4B* and *Supplementary file 1*). This high density of cross-links leads to a precise localization of the first half of Med1, with Cluster 1 showing low root-mean-square fluctuations (RMSFs) for residues within this portion of the protein. In contrast, the C-terminus of Med1 lacks cross-links (*Figures 1D, 4B*) and is restrained only by the EM density, resulting in a lower precision of the model and higher RMSF values (*Figure 2—figure supplement 5*).

## Tail module architecture

The region of the Tail module that abuts the Middle module is a rich subunit interaction hub, composed of residues from the N-termini of Med2 and Med3 and the C-termini of Med15 and Med16 (*Figures 3, 5, 6C*). The C-terminus of Med14 also contributes to the structure of this region, consistent with the finding that a C-terminal deletion of Med14 results in the loss of all Tail subunits from Mediator (*Li et al., 1995*). The Med2 and Med3 NTDs are modeled with relatively high precision (*Figure 2—figure supplement 5*), and structural similarity searches with these NTDs show a number of high similarity hits to coiled-coil proteins, in particular to multiple chains within the highly elongated Fibrinogen trimeric coiled-coil structure (*Figure 5—source data 1*). We extended this analysis to calculate the odds that each of these NTDs forms a coiled-coil structure, with the use of two

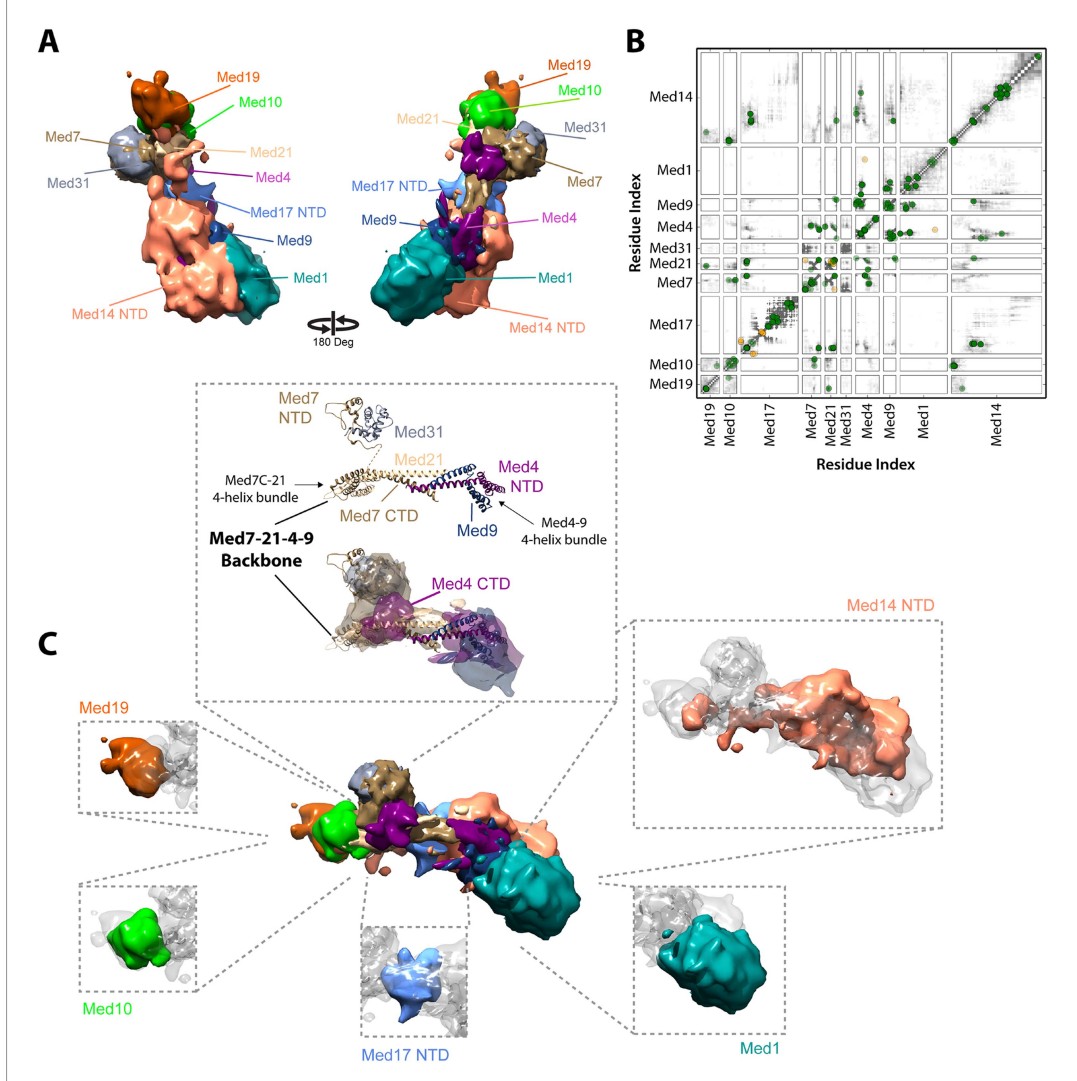

**Figure 4**. Architecture of the Mediator Middle module. (**A**) Blow-out views of the Middle module subunit localization maps with the first view and coloring identical to *Figure 2A* and the second view related by a 180° rotation around the y-axis. (**B**) Average contact maps calculated for the cluster of best scoring solutions. Each square is a contact map calculated between a given pair of Middle module proteins with border length proportional to the length of corresponding subunit sequences. The grey-shaded areas indicate observed interactions, with the grey-scale proportional to the fraction of models observing the corresponding interaction. The colored circles are observed cross-links, where the green and orange colors represent respectively satisfied and violated cross-links for the cluster center (i.e., the solution that has the minimal root-mean-square deviation (RMSD) from all the other solutions in the cluster). (**C**) Top view of Middle module with individual panels showing the full localization map of each Middle subunit within its surrounding density, made semi-transparent for visual clarity. In the central portion of the Middle module the end-to-end stacking of Med7C-21 (PDB 1YKH) and Med4-9 (Homology Model [*Lariviere et al., 2013*]) heterodimers forms a backbone scaffold (top panel), upon which Med10 and Med19 associate at the Med7C-21 extreme (left panels) while Med1 associates closely with the N-terminal portions of Med4-9 (bottom right panel). The Med7N-31 complex (PDB 3FBI) is located proximal to the Med7C-21 heterodimer and the unmodeled carboxy-terminal domain (CTD) of Med4 (top panel). The centrally located Med17 NTD (bottom panel) is wedged between the Med7C-21-4-9 backbone and Med14, which forms contacts with Med10 and Med21 at its extreme NTD, while the bulk of its density localizes to Med4-9 and Med1 (top right panel). Subunit localization density and ribbon models are colored according to *Figure 2A*.

independent prediction servers (*Figure 5D*). We found a good correspondence between regions of similarity to coiled-coil proteins and regions predicted with high confidence (p > 0.8) to form coiled-coil structures. These analyses, together with the co-localization of these domains in our

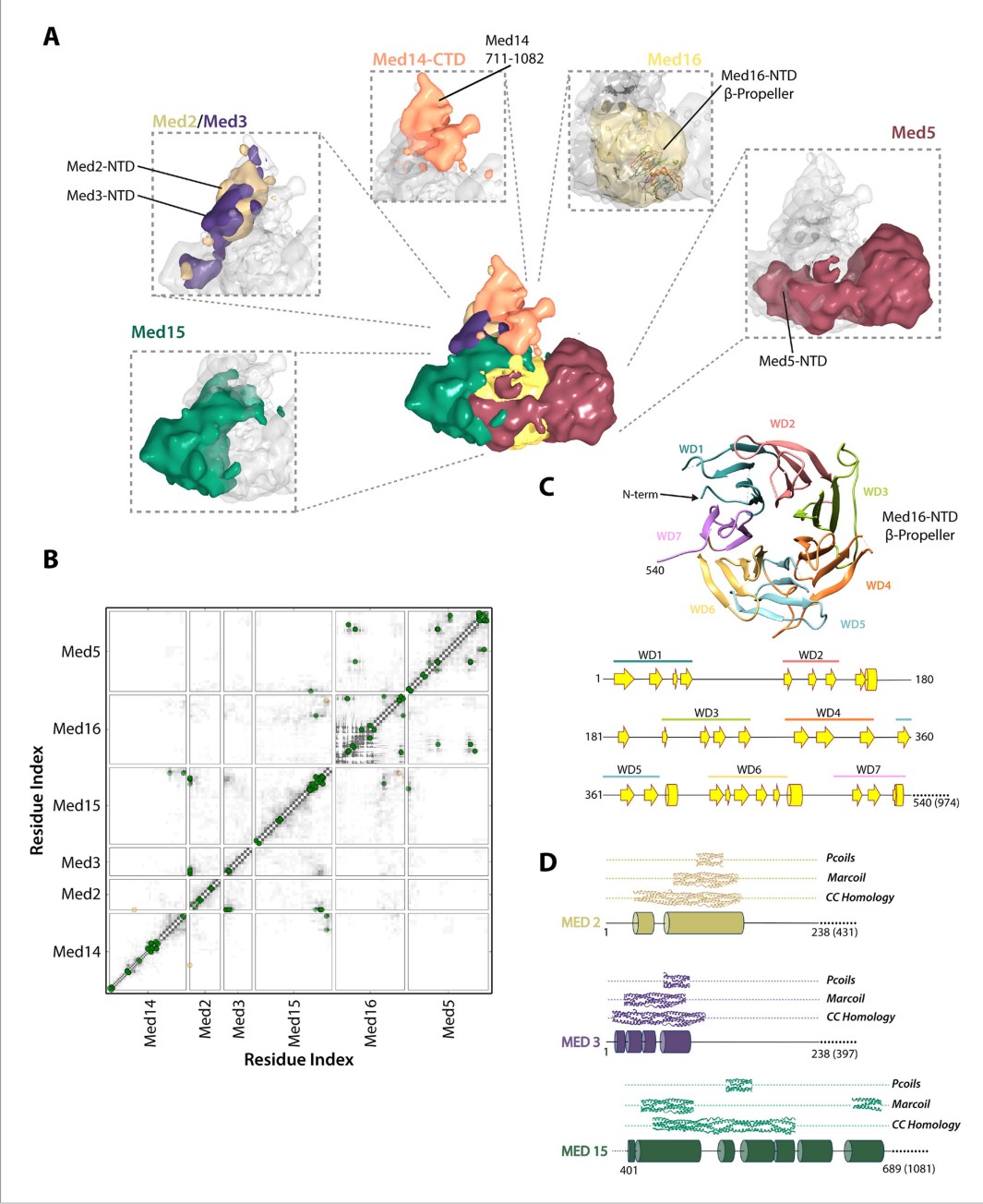

**Figure 5**. Architecture of the Mediator Tail module. (**A**) Subunit localization within the Tail module. For each Tail subunit, the corresponding localization density is shown within a semi-transparent Tail density (grey) and colored according to *Figure 2A*. (**B**) Average contact maps calculated for the cluster of best-scoring solutions and represented as described in *Figure 4B*. (**C**) β-propeller model of the Med16 NTD. The location of 7 WD domains within the N-terminus of Med16, predicted by MSA analysis (*Bourbon, 2008*), is shown in the context of the predicted Med16 N-term secondary structure (bottom panel). Similarity searches identified a high-confidence match to the 7-bladed β-propeller of the *S. cerevisiae* vesicle coat protein Sec31 (PDB 2PM9). (**D**) Schematic showing the predicted coiled-coil regions of three interacting Tail proteins, Med2, Med3 and Med15. The co-localization of the Med2/3 N-termini and Med15 C-term (panel **A**) supports the formation of a coiled-coil at this tail locus.

The following source data is available for figure 5:

**Source data 1**. HHpred comparative modeling results for Tail module proteins.

Mediator model and the previous observation that Med2 and Med3 can be isolated as a heterodimer (*Beve et al., 2005*), suggest that the N-termini of Med2 and 3 form a coiled-coil motif in the Middle-Tail contact region.

Med15 is co-localized in our Mediator model with Med2 and Med3 and can be isolated with Med2 and Med3 in a trimeric complex (*Zhang et al., 2004*). Central residues of Med15 exhibit homology to Fibrinogen (*Figure 5—source data 1*), and may participate in a coiled-coil structure, but it is the C-terminal domain of Med15 (850–1081) that interacts with Med2 and Med3 in the Middle-Tail contact region. The Med15 CTD also interacts with the Med16 CTD, apparently stabilizing the association of Med16 within the Tail, and that of Med5 as well, because truncation of the Med16 CTD results in the dissociation of both Med16 and Med5 from the Mediator (*Tsai et al., 2014*).

Previous sequence analysis of Mediator subunits identified seven WD repeats in the N-terminal domain of Med16 (*Figure 5C*) (*Bourbon, 2008*). We extended this analysis and identified a number of high-confidence hits to seven-bladed β-propeller structures. The most complete match to a full β-propeller was to Sec31, a component of the *S. cerevisiae* coat protein complex (COPII) involved in ER vesicle transport (*Figure 5C* and *Figure 5—source data 1*). We used a homology model, in which the first 540 residues of Med16 were mapped onto the Sec31 β-propeller structure, in our modeling of the Tail module architecture (*Figure 2—figure supplement 2*). Two intra-protein cross-links of Med16 could be mapped onto the homology model without violation (*Supplementary file 1*). In our model, the Med16 β-propeller was located in a central position at the base of the Tail module, making extensive contacts with Med5 (*Figure 6D*). The central location of Med16 within the Tail module and a large interaction surface with Med5 are consistent with the finding that the entire Tail module is destabilized and lost in Mediator preparations from a strain from which the *MED16* gene was deleted (*Li et al., 1995*). In contrast, Med5 occupies a terminal position within the Tail module, and the *MED5* gene can be deleted without disrupting the remainder of the module (*Tsai et al., 2014*). Although the majority of Med5 occupies a single location within the Tail module, the N-domain of Med5 follows a path around the outside edge of the Med16 β-propeller domain before interacting with Med15 at a more internal locus (*Figure 6D*). This interaction is consistent with an earlier finding that Med5 can form a tetrameric complex with the Med2-Med3-Med15 trimer in the absence of Med16 (*Beve et al., 2005*).

## Unique architectural roles of Med17 and Med14

Especially noteworthy in our Mediator model are the special roles performed by Med17 and Med14. An integral component of the Mediator Head module, Med17 is of particular interest for a temperature sensitive allele (*srb4-138^ts*) in which all pol II-dependent transcription is abolished at the restrictive temperature (*Thompson and Young, 1995*; *Holstege et al., 1998*). N-terminal truncations of Med17 in *S. cerevisiae* are inviable (data not shown). The N-terminal 181 residues of Med17 are disordered in Mediator Head module crystals. Our Mediator model reveals a central role of the Med17 N-terminal domain in the connection between the Head and Middle modules. The Med17 NTD is strongly localized at a central site in the Middle module, wedged between Med7-Med21-Med4-Med9 backbone density on one side and Med14 density on the other (*Figures 2, 4*). The entire path of Med17 from this Middle interaction site to its first modeled residue in the Head module is clearly resolved in the model (*Figure 6A*).

The essential Med14 subunit was originally identified as a repressor protein in yeast (*Sakai et al., 1990*). Reports of the location of Med14 in the Mediator have been contradictory. Med14 truncations uncouple the Tail module, suggesting that Med14 is primarily a Middle module protein (*Li et al., 1995*), but in yeast strains in which the Middle module has been disrupted by subunit deletion, Med14 remains associated with the Tail (*Baidoobonso et al., 2007*). Furthermore, Med14 is required along with reconstituted Head and Middle modules for basal transcription in vitro in Mediator-depleted cell extracts (*Cevher et al., 2014*). Med14 formed cross-links with components of the reconstituted Head and Middle modules, as well as with Med17 (*Cevher et al., 2014*).

In our Mediator model, Med14 makes extensive contacts with proteins from all three modules, and is the only Mediator subunit that does so (*Figure 1D*). It spans almost the entire Mediator complex, extending a distance of about 220 Å from N-terminal interactions with Med10-Med19 at the tip of the Middle module to C-terminal interactions at the Middle-Tail contact region (*Figure 6B,C*). Med14 residues 264–600 interact with the Med4-Med9 four-helix bundle (*Figures 3, 4, 6B*). Med14 residues 247–350 participate in bridging the Head and Middle modules through multiple contacts with the

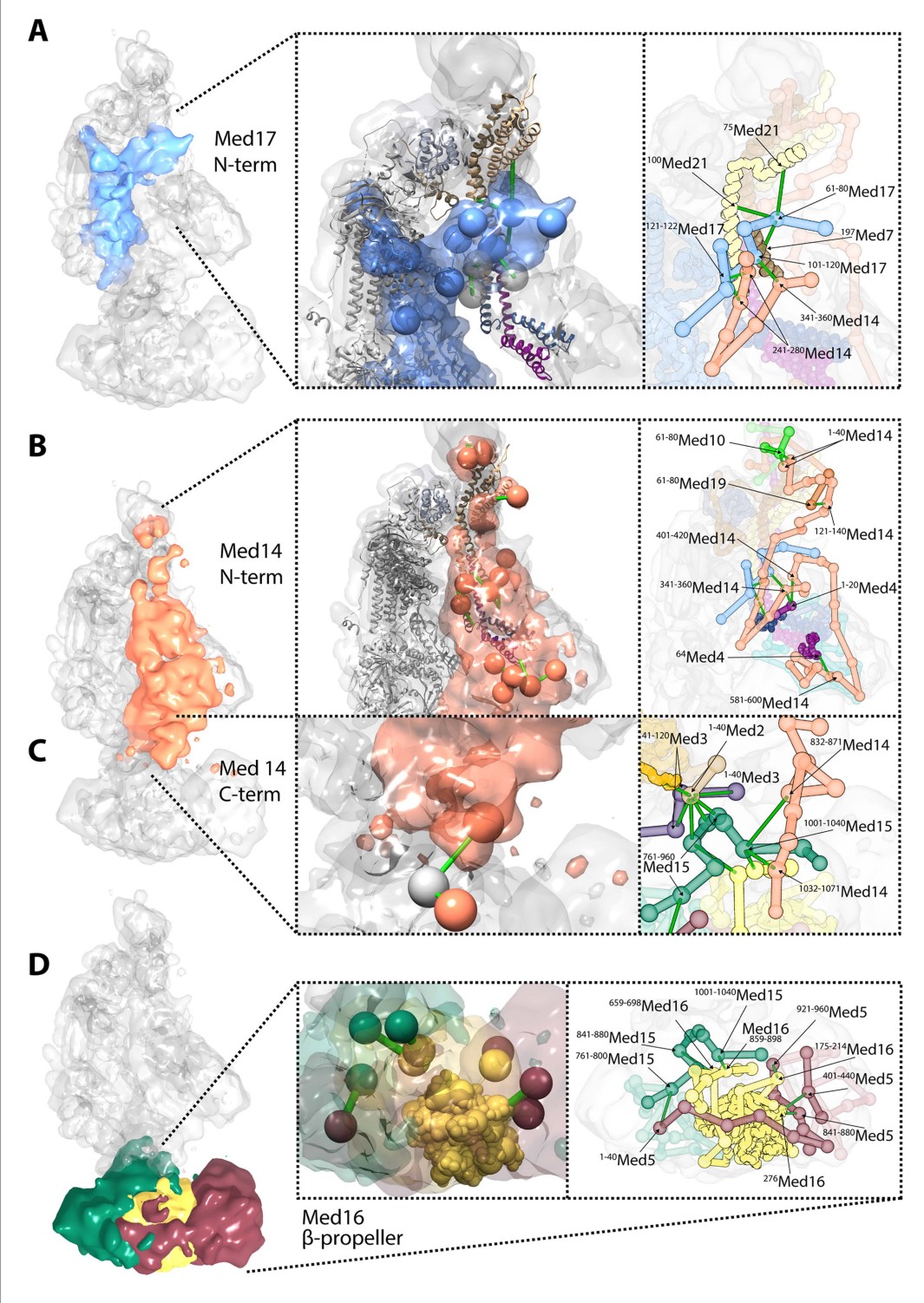

**Figure 6**. Novel structural insights into Mediator complex architecture and module connectivity. (**A**) Subunit localization density for Med17 (blue) reveals the extension of the unmodeled N-terminal domain to contact Middle module subunits. (Left **A** panel) The N-terminus of the Head subunit Med17 (blue density and beads) forms an extensive cross-linking network (green links) within the Middle module, acting as an inter-modular bridge (see right **A** panel and *Supplementary file 1* for detailed cross-link information). Head and Middle module ribbon models are colored light grey and according to their subunit color in *Figure 2A*, respectively, and unmodeled residues cross-linked to Med17 are represented with light grey beads. Only Med17 beads with cross-links are shown for clarity and

*Figure 6. continued on next page*

*Figure 6. Continued*

coordinates are derived from the full-complex centroid model. (Right **A** panel) A detailed stick representation of the cross-linking network shown in the left panel. (**B**) Subunit localization density for Med14 (salmon) highlights its unique function as a Mediator scaffold protein spanning ~220 Å to connect all three modules. (Left **B** panel) The N-terminus of Med14 forms an extensive cross-linking network across the full Middle module. The structural elements of Med14 and other subunits are represented as in left panel of (**A**). (Right **B** panel) A detailed stick representation of the cross-linking network shown in left panel. (**C**) The localization of the Med14 CTD to the Middle-Tail junction is observed through a pair of cross-links to a Tail protein located in the region (Left **C** panel). (Right **C** panel) A detailed stick representation of the subunit localization and cross-linking network within the Middle-Tail junction. (**D**) Subunit localization density for the subunits localized in the central portion of the Tail module. (Left **D** panel) Bead representation of the cross-links involving the N- and C-terminal portions of Med16 (yellow) with Med5 (brown) and Med15 (green), respectively. The Med16 N-terminal β-propeller is centrally located in the Tail module and forms extensive interactions with Med5. The Med16 CTD links Med16-Med5 with the remainder of the Tail. (Right **D** panel) A detailed stick representation of the cross-linking network shown in left panel. Coordinates for both panels of (**C**) and (**D**) are derived for the Tail module centroid model.

Med17 NTD. Additional interactions with the Head module proteins are suggested by contacts of Med14 residues 601–1082 with Med20 and the C-terminal region of Med17 (residues 401–687) (*Figure 3*).

## Holoenzyme configuration

*Plaschka et al. (2015)* recently reported a 9.7 Å structure of a yeast 'core initiation complex', comprising the Mediator Head module, a minimal Middle module, pol II, a nucleic acid scaffold, and the general transcription factors TBP, TFIIB and TFIIF, obtained by cryo-EM and chemical cross-linking (*Plaschka et al, 2015*). The Mediator portion of the cryo-EM map represents only about 50% of the mass of Mediator, as the 'core initiation complex' does not include the 480 kDa Tail module or the Middle module protein Med1. The structure shows the 'neck' of the Head module binding to pol II near subunits Rpb 4 and Rpb7, with the 'mobile jaw' regions of Med 18 and Med20 extending to contact the TFIIB-binding site and subunits Rpb 3, 10, 11, and 12. Cross-links between subunits within the Head (30 unique cross-links) and Middle (71 unique cross-links) modules are in complete agreement with our three-dimensional architectural model of the free Mediator complex (*Figure 7—figure supplement 1B*). Docking our Mediator model to the EM map of *Plaschka et al. (2015)* resulted in a holoenzyme model that was also largely consistent with Mediator to polymerase cross-links (12 non-redundant and within the modeled sequence) identified by *Plaschka et al. (2015)* (*Figure 7A,C*). However, a set of 17 Mediator to polymerase cross-links from our study of holoenzyme showed agreement only in the region where the Head module contacts Rpb4 and Rpb7 (*Figure 7B*). Cross-links between the Tail and Middle modules to pol II in our study were not consistent with the map of *Plaschka et al. (2015)* (*Figure 7C*). As mentioned above, cross-links between Mediator modules in our study were also poorly satisfied by the EM density for free Mediator used in our integrative modeling. Together, these findings suggest a different conformation of the Mediator-polymerase holoenzyme in the presence of the Tail module (not included in the map of *Plaschka et al. (2015)*) and in the absence of the nucleic acid scaffold and general factors. Our results point to a similar contact site between the Head module and pol II to that in the model of the 'core initiation complex', but suggest that a conformational change in the Mediator brings the Tail module in contact with pol II near the DNA binding cleft of pol II in the holoenzyme, a suggestion that is consistent with our own preliminary cryo-EM data on the holoenzyme (P Robinson, unpublished) (*Figure 7D*). Such conformational dynamics are in keeping with the idea that interaction between Mediator and pol II in the absence of other factors is largely determined by interactions between the Rpb1 CTD and the neck of the Head module (*Robinson et al., 2012*).

## Discussion

Through integration of data from diverse sources, we have arrived at a 3-D model of Mediator, which explains virtually all previous findings, and which provides a complete picture, including internal details, of

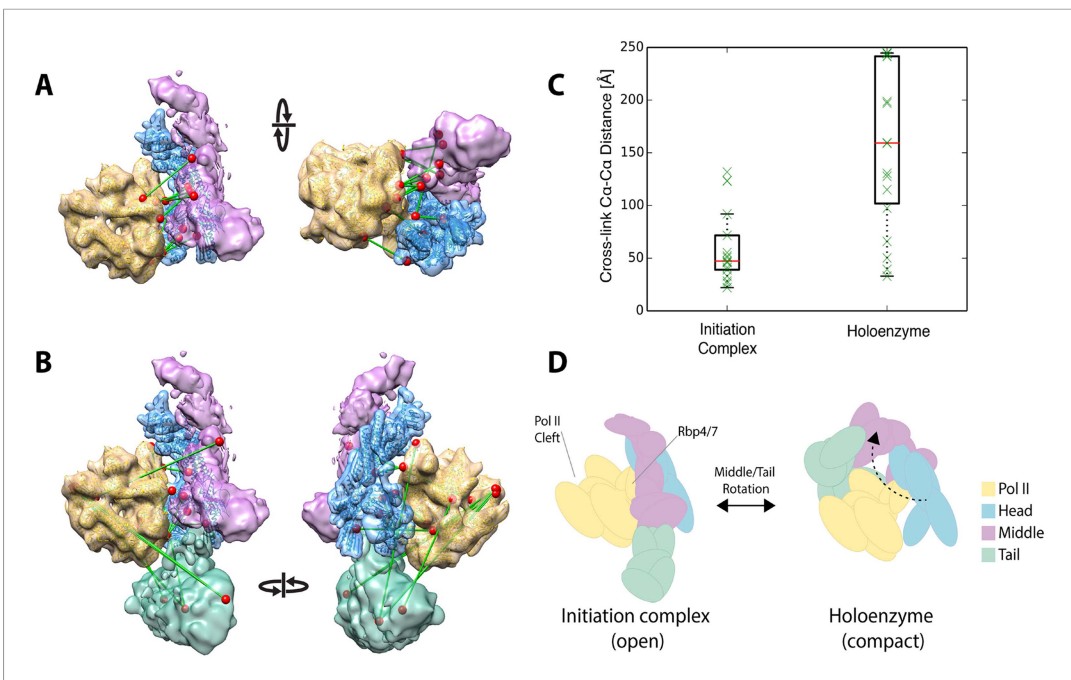

**Figure 7**. Holoenzyme cross-linking data indicate different conformational states between the holoenzyme and the core Mediator initiation complex. (**A**) Docking of our Mediator model to the EM density of *Plaschka et al. (2015)* provides a subunit architectural map that is highly consistent with the 12 Mediator to pol II cross-links of *Plaschka et al. (2015)* (non-redundant and within modeled sequence), collected on a core initiation complex containing the Mediator Head and Middle modules with pol II, nucleic acid scaffold and the general factors TBP, TFIIB and TFIIF. (**B**) In contrast, the 17 equivalent Mediator to pol II cross-links of the current study are largely inconsistent with the position of Mediator modules found in the core initiation complex. (**C**) Distributions of cross-link Cα distances from the two datasets measured in the context of the docked Mediator model. (**D**) The pattern of holoenzyme cross-links implies a major conformational rearrangement in the presence of the Tail module and absence of DNA scaffold and general transcription factors.

The following figure supplement is available for figure 7:

**Figure supplement 1**. Further validation of the Mediator model.

the complex. One source of data was cross-linking, performed on a native holoenzyme, comprising pol II and Mediator, for several reasons: Mediator structure was thereby analyzed in a more functionally relevant state; the native holoenzyme, isolated from yeast, was much more soluble and stable than a reconstituted holoenzyme formed by mixing pol II and Mediator in vitro; and the X-ray crystal structure of the pol II component of the holoenzyme provided an internal control for the modeling process. We restricted our analysis to the internal organization of the three Mediator modules, because their relationship to one another is believed to differ between free Mediator and the holoenzyme, and free Mediator was the source of our EM information. Notable features of our 3-D Mediator model include the following:

1. The definition of the three Mediator modules. Whereas Med17 and Med14 were previously classified as Head and Tail proteins, the N-terminal domains of both proteins are now seen to form part of the Middle module. The two NTDs not only interact with core components of the Middle module but also with each other.
2. The special roles of Med17 and Med14 in the architecture of the entire Mediator. Med17 connects Head and Middle modules, while Med14 extends the entire length of Mediator, connecting all three modules. The path of the NTD of Med17, not revealed in X-ray crystal structures of the Head module, can now be followed from a site of interaction in the Middle module to its connection with the body of the protein in the Head module.
3. The complete 3-D arrangement of subunits of the Middle module. A Med7-Med21-Med4-Med9 tetramer is seen to form the core of the module, with a cap of Med10-Med19 at one end and a cap of Med1 at the other.

4. The complete 3-D arrangement of components of the Tail module. Of particular note are the extensive interactions of the NTDs of Med2 and Med3 and the CTDs of Med 15 and Med16, which form the region of the Tail module that contacts the Middle module, and which is important for Mediator-activator protein interaction. The Med2-Med3-Med15 triad is a binding target of multiple yeast transcriptional activator proteins, including Gcn4 and Gal4 (*Lee et al., 1999*; *Myers et al., 1999*; *Zhang et al., 2004*). Yeast harboring a Med3Δ Med15Δ double knockout exhibits decreased levels of Mediator and of transcription pre-initiation complexes at inducible gene promoters (*Ansari et al., 2012*).

5. Secondary structural features of Tail module proteins. The NTDs of Med2 and Med3 interact in a coiled-coil motif in the Middle-Tail junction. A 7-bladed β-propeller encompassing the N-terminal half of Med16 interacts extensively with Med5 at the base of the Tail. Both proteins play roles in transcriptional repression, and expression analysis in yeast Tail mutants has established a functional link between them (*van de Peppel et al., 2005*).

The 3-D Mediator model not only unifies and resolves discrepancies in the extensive literature on Mediator organization, but also forms a starting point for future investigation. High-resolution structures can be docked and additional structural information incorporated in the model, using the same integrative approach we applied here (*Russel et al., 2012*). Interactions with pol II, general transcription factors, and transcriptional regulatory proteins can be mapped to the surface. Changes associated with activity or due to mutation can be identified and their functional significance understood.

Our Mediator model could be docked into the Mediator portion of the EM map of *Plaschka et al. (2015)*, as association with pol II appears to cause little change in the structure and relative position of the Head and Middle Mediator modules compared to those in the free Mediator EM map used as restraint in our modeling study (*Tsai et al., 2014*) (*Figure 7—figure supplement 1A*). Our docked model provides a more detailed map of the locations of Middle module subunits across the pol II surface, and therefore allows a more complete interpretation of the Mediator portion of the map of *Plaschka et al. (2015)*. Our model provides a structural basis for the Mediator to pol II cross-links reported by *Plaschka et al. (2015)*, which can be mapped with precision onto both the Mediator and pol II surfaces to yield a structurally consistent set of cross-link distances (*Figure 7A,C*). Furthermore, mapping novel Middle module cross-links from *Plaschka et al. (2015)* onto our Mediator model, we find additional strong support for the validity of our subunit localizations (*Figure 7—figure supplement 1B*).

Also as mentioned above, the cross-links between Mediator modules that we obtained with holoenzyme are inconsistent with the result of docking our Mediator model into the map of *Plaschka et al. (2015)* (*Figure 7B,C*). We found multiple cross-links between the Tail module and the region of pol II around the active center cleft, whereas docking our Mediator model into the map of *Plaschka et al. (2015)* places the Tail module on the opposite side of pol II. We have noted the Mediator conformational change required to satisfy our Tail module-pol II cross-links (*Figure 7D*). Tail module interactions with pol II in the vicinity of the active center cleft may be relevant to the role of Tail module subunits Med14, 15 and 16 in transcriptional repression at multiple loci in yeast (*Covitz et al., 1994*; *Sussel et al., 1995*).

The success of the integrative modeling approach in the derivation of Middle and Tail module structures warrants application to the issues raised here and to other challenging problems. It is of immediate interest to apply IMP to the holoenzyme with an expanded cross-link dataset and a high resolution holoenzyme EM map. Further studies may be directed towards a complete RNA polymerase II pre-initiation complex including Mediator, and to additional biological particles of great size and complexity.

## Materials and methods

### Yeast strains

The *S. cerevisiae* Holoenzyme was purified as an intact complex using a yeast strain in which both endogenous Mediator and RNA Pol II were affinity tagged. To achieve double tagging the following genetic manipulations were performed in the CB010 background (*Matα pep4::HIS3 prb1::LEU2 prc1::HISG can1 ade2 trp1 ura3 his3 leu2–3 leu2–112*). First, the N-terminus of the essential Mediator Head module subunit, Med17p, was TAP-tagged using the N-terminus tagging cassette, including

*Kluyveromyces lactis TRP1* marker gene, amplified from vector pBS1761 as described previously (*Puig et al., 2001*). The *MED17* gene was returned to endogenous promoter regulation following Cre-mediated LoxP recombination driven from pSH47 (Ura+). The C-terminus of the largest RNA Pol II subunit, Rpb1p, was tagged with a custom protein G affinity tag with 3C PreScission protease cleavage site. The vector for C-terminal protein G tag cassette amplification [pCEMM-CTAP(KanMX)] was first generated by sub-cloning a 1633 bp fragment containing the Tn903 *KanMX* marker gene with flanking LoxP sites from pUG6 between the XhoI (597) and NotI (616) sites of mammalian expression vector pCEMM-CTAP(SG), positioned immediately C-terminal to the protein G domain sequence (Forward primer 5′-3′: actgtcactgaatagctcgaggaacgcggccgccagctgaagcttcgtacg–Reverse primer 5′-3′: ggcggaatttacgtagcggccgcccgcggccgcataggccactagtggatctg). The Rpb1 C-terminal protein G tagging cassette including the protein G and KanMX elements was PCR amplified from pCEMM-CTAP (KanMX) using primers with flanking sequence including 45 bp complementary to Rpb1 genomic DNA immediately upstream/downstream of the STOP codon and sequence encoding the 3C PreScission cleavage site (5′-3′: ctggaagttctgttccaaggtcca).

## Native *S. cerevisiae* holoenzyme complex protein purification

The endogenous yeast complex was purified from whole-cell extracts using an affinity capture method described previously with minor modifications (*Robinson et al., 2012*). Double-tagged CB010 were grown in 200L YPAD to 9.0 $A_{600}$ and 3.0 Kg cells were lysed by continuous-flow bead beating in A25 buffer (25 mM ammonium sulfate, 100 mM HEPES pH 7.8, 1 mM EDTA, 5% glycerol, 5 mM DTT, and 1× protease inhibitor cocktail [0.6 µM leupeptin hemisulfate, 2 µM pepstatin A, 1 mM PMSF and 2.1 mM benzamidine hydrochloride]) with 25 µg/ml RNAse A (Qiagen, Hilden, Germany). The cell debris was pelleted by centrifugation at 12,250×$g$ for 1 hr at 4℃ and the supernatant made up to 300 mM ammonium sulfate with the addition of cold 3.9 M ammonium sulfate and gentle stirring. Nucleic acids were depleted from the lysate by adding 500 ml DEAE (GE healthcare, Little Chalfont, UK) pre-equilibrated in A300 buffer. After 30 min of stirring the DEAE was pelleted by centrifugation (12,250×$g$, 30 min at 4℃) and the supernatant loaded onto an IgG column before washing in A500 buffer. The use of two distinct protease cleavage sites allowed the isolation of stable holoenzyme complex using a two-stage elution approach (*Figure 1—figure supplement 1*). First-stage TEV cleavage led to the release of free Mediator, with Mediator retention occurring through complex formation with uncleaved RNA pol II. Subsequent cleavage with 3C PreScission protease led to the release of both free RNA pol II and holoenzyme complex. The order of protease cleavage could be reversed with equivalent yields of holoenzyme complex (∼5 mg/3 Kg dry cell mass). The holoenzyme was further purified from free RNA pol II (TEV-3C) or Mediator (3C-TEV) using ion exchange (75–600 mM $(NH_4)_2SO_4$; HiTrap FFQ) and size-exclusion (200 mM $(NH_4)_2SO_4$, 25 mM HEPES, pH 7.8, 5% glycerol, 2 mM DTT; Superose 6) chromatography steps. In the case of the TEV-3C cleavage order, the holoenzyme eluted from the IE column as a leading shoulder in the large peak due to free RNA pol II. Using the 3C-TEV cleavage order, the holoenzyme was retained longer in IE than free Mediator and could be resolved as an independent peak.

## Cross-linking procedures

Prior to cross-linking, holoenzyme samples were exchanged into phosphate buffer containing: 150 mM $NaPO_4$, 150 mM KOAc, pH 7.5, 5% glycerol and 2 mM DTT by dialyzing against three changes of buffer. Cross-linking was then performed by one of two procedures:

### Procedure A

A 50:50 mixture of unlabeled and $^2H_{12}$-labeled BS3 reagent (Creative Molecules, Victoria, BC, Canada) was applied to generate cross-linked reaction products spaced by 12 Da. Approximately 315 µg holoenzyme was treated with 2.7 mM reagent (120 min on ice) and then quenched with ammonium sulfate. Mono-dispersed Holoenzyme was separated from aggregate by size-exclusion chromatography (Superose 6). The cross-linked holoenzyme (estimated at 160 µg) was then treated with 0.8 M Urea, 10 mM TCEP (20 min at 56℃), alkylated with 20 mM iodoacetamide (1 hr at room temperature), adjusted to pH 8 with 100 mM ammonium bicarbonate, and digested with trypsin (Promega, Fitchburg, WI, United States) added at a 1:40 ratio (overnight at 37℃). The digests were acidified to 0.3% TFA and peptides were solid-phase extracted using a 100 µl $C_{18}$ OMIX tip (Agilent, Santa Clara, CA, United States). Digests were fractionated by high pH $C_{18}$ chromatography using a

1.0 × 100-mm column packed with 3 µm, 110 Å Gemini $C_{18}$ resin (Phenomenex, Torrance, CA, United States) (Mobile phases A: 10 mM ammonium formate, pH 10; B: 10 mM ammonium formate, pH 10 in 50% ACN) (*Robinson et al., 2012*).

Both fractionated and unfractionated samples were analyzed on an LTQ-Orbitrap Velos (Thermo Scientific, San Jose, CA, United States) mass spectrometer equipped with a Nanoacquity UPLC system (Waters, Milford, MA, United States). Peptides were re-suspended in solvent A (0.1% formic acid (aq)) and separated at 600 nl/min on a 75 µm × 150 mm BEH130 C18 column (Waters). Gradients of varying lengths were run from 3-27% solvent B (0.1% formic acid in ACN) followed by a wash at 50% B. Precursor ion spectra were measured in the Orbitrap at 30,000 resolution while HCD product ions were measured in the Orbitrap analyzer at 7500 resolution. Half of each sample was acquired using a data-dependent protocol in which the six most abundant triply-charged or higher ion signals in each precursor scan were selected for HCD. A dynamic exclusion window was used to prevent re-selection of the same precursor ion. These data were processed offline with in-house scripts to identify precursor ion signals with the characteristic monoisotopic spacing of 12.0753 Da. The remainder of each sample was then re-acquired using an inclusion list protocol, which only targeted these precursor ions (both light and heavy pairs were targeted). Peaklists from both sets of acquisitions were later merged and searched for cross-linked products.

### Procedure B
Approximately 380 µg of holoenzyme was cross-linked with BS3 (Pierce, Rockford, IL, United States) in phosphate buffer and quenched with ammonium sulfate. This sample was directly treated with 0.8 M Urea, 10 mM TCEP at 50°C for 25 min and otherwise digested as in Procedure A. Desalting was performed using a Peptide Macrotrap $C_{18}$ cartridge (Bruker, Billerica, MA, United States). Peptides were loaded at 250 µl/min in 0.3% TFA (aq) and eluted at 500 µl/min in 70% ACN, 0.3% TFA (aq). Desalted peptides were then enriched for cross-linked products using a Superdex Peptide PC 3.2 × 300 mm column (GE Healthcare) as described previously (*Leitner et al., 2012*). All chromatographic steps were performed on an Akta purifier system (GE Healthcare). Mass spectrometry was performed as in procedure A for data-dependent acquisition.

## Interpretation of cross-linking-MS data
MS Peaklists were generated such that all monoisotopic ions detected within the selection window of the linear ion trap (4 m/z units) were annotated as possible precursors to a given product ion spectrum, as described previously (*Robinson et al., 2012*; *Trnka et al., 2014*). Peaklists were first searched against the full SwissProt database to check the composition of the sample. Cross-link searches were then carried out against a target database consisting of the 52 components of the *S. cerevisiae* Preinitiation Complex. For each polypeptide sequence, 10 randomized versions were generated and concatenated to the target database for a total of 572 sequences (52 target and 520 decoy). Decoy protein sequences were the same length as their corresponding target sequences and sampled from the natural distribution of *S. cerevisiae* amino acid frequency.

Peaklists were then searched against this database using Protein Prospector (version 5.13.1) to assign putative cross-linked peptides (*Trnka et al., 2014*). Carbamidomethylation of cysteine was considered as a fixed modification. N-terminal methionine loss with and without acetylation, peptide N-terminal glutamine conversion to pyroglutamate, oxidation of methionine, and 'dead-end' modification of lysine and the protein N-terminus by semi-hydrolyzed BS3 were considered as variable modifications in addition to cross-linking by BS3. Up to 3 variable modifications per peptide were considered. Mass tolerances of 13 ppm and 20 ppm were used for precursor and product ions respectively. Trypsin specificity with 3 missed cleavages was used to generate theoretical peptides. 85 product ion signals from each MS2 spectrum were used for assessment. For experiments involving isotopic mixtures of cross-linking reagents, the searches were performed twice, once specifying heavy BS3 and once with light BS3. The search results were later merged. Both light and heavy dead-end modifications were considered in each of these database searches.

For all experiments, cross-link spectral matches (CSM) were discarded if the following Protein Prospector parameters fell outside the threshold values: peptide score below 20, peptide/protein/worse_peptide expectation values above 50, and score_difference below 0. A linear support vector machine (SVM) model was constructed to classify CSMs between decoy and target classes (*Trnka et al., 2014*). Briefly, the CSMs were split evenly into two sets. SVM models were trained on half of the data and evaluated for their ability to correctly classify true negatives (hits to the decoy database) on

the test set. CSMs were considered decoy matches if either of the two constituent peptides came from the decoy database. The final SVM model included a metric reflecting the number of product ions matched to the less fragmented of the two peptides (score_difference) and a metric for the overall match of the spectrum (percentage of the total ion current annotated). The inclusion of additional metrics did not increase the predictive power of the model. An SVM decision value of 0 was taken as the acceptance threshold for reporting cross-linked peptides. FDRs were calculated by dividing the number of decoy hits at this threshold by 10 (to account for the larger decoy database size) and dividing by the number of hits to the target database.

CSMs where either peptide was less than 4-amino acids long were discarded, with a few manually validated exceptions. When spectra could be interpreted by multiple cross-linked peptide-pairs, CSMs were considered to be unambiguous if the second best match was further than 0.3 SVM decision value units from the top match. Intra-protein cross-links were selected over inter-protein cross-links if there were still conflicts. The remaining decoy matches were also removed from consideration. Following these selection criteria, most CSMs identified a single cross-linked peptide-pair. The remaining spectral redundancy came exclusively from ambiguous site localizations, where there was sufficient evidence to identify both peptides, but the position of the modified amino acid remained ambiguous. In these cases, all possible sites were reported using a spectral identifier and used in subsequent modelling steps.

Cross-linking results and annotated HCD spectra may be viewed online using Protein Prospector's MS-Viewer program: http://prospector2.ucsf.edu/prospector/cgi-bin/msform.cgi?=msviewer.

Search key: qzpxihtngx

All raw mass spectrometry files and peaklists used in the study have been deposited to the MassIVE proteomics repository and are available through the ProteomeXchange consortium.

MassIVE accession: MSV000079237

ProteomeXchange accession: PXD002723

## Integrative structure modeling

Our integrative structure modeling of RNA Pol II and Mediator complexes proceeds through four stages (*Lasker et al., 2010*; *Fernandez-Martinez et al., 2012*; *Lasker et al., 2012*) (*Figure 2*): (1) gathering of data, (2) representation of subunits and translation of the data into spatial restraints, (3) configurational sampling to produce an ensemble of models that satisfies the restraints, and (4) analysis and assessment of the ensemble. The modelling protocol (i.e., stages 2, 3, and 4) was scripted using the Python Modelling Interface (https://github.com/salilab/pmi), version be72c15, a library to model macromolecular complexes based on our open source IMP package (http://salilab.org/imp/), version 829c3f0 (*Russel et al., 2012*). Files containing the input data, scripts, and output models are available at http://salilab.org/mediator.

## Molecular architecture of the Mediator complex revealed by integrative modeling

### Stage 1: gathering of data

The current study identified 260 BS3 unique cross-linked positional-pairs ('cross-links') corresponding to the Mediator complex (*Figure 1D*, *Figure 1—source data 1*). This dataset was supplemented by 38 unique cross-links (filtered by our acceptance criteria where applicable) from a previous independent study of reconstituted Middle module (*Lariviere et al., 2013*). The experiment of Larivière et al. utilized the CBDPS reagent, which is a few Å longer than BS3 and also lysine directed. There were 4 cross-links in common between the datasets producing 294 cross-links in the combined data. These cross-links were identified by a total of 1900 spectral matches, which consist of multiple peptide sequences, charge states, or replicates.

The atomic structures for the full yeast Mediator Head module (80% sequence coverage) as well as the heterodimers of $^{111\text{-}206}$Med7-$^{1\text{-}130}$Med21 (PDB 1YKH) and $^{12\text{-}84}$Med7-$^{19\text{-}110}$Med31 (PDB 3FBI) in the Middle module have been previously determined by X-ray crystallography (*Figure 2—figure supplement 2*) (*Baumli et al., 2005*; *Koschubs et al., 2009*; *Robinson et al., 2012*). Overall, the crystallographic structures covered 23% of the residues in the Mediator complex. In addition, a comparative model of the Middle module heterodimer $^{37\text{-}127}$Med4-$^{65\text{-}149}$Med9 was previously constructed based on the structure of the $^{111\text{-}206}$Med7-$^{1\text{-}130}$Med21 (*Lariviere et al., 2013*).

A comparative model of the N-terminal portion of Med16 containing 7 WD motifs (8–538) was discovered and initially computed using the Phyre2 server (*Figure 5*) (*Kelley and Sternberg, 2009*). The model was further refined by manually adjusting the sequence numbering and removing large, loop regions. Regions with poor stereochemistry were adjusted using the Coot structural refinement tools (*Emsley et al., 2010*).

## Exhaustive docking of head module within EM density

Finally, an 18 Å resolution cryo-EM density map of the Mediator complex was recently determined (EMD- 2634, 25 Å resolution using FSC 0.5) (*Tsai et al., 2014*), which is currently the highest resolution map available. We used an unbiased, exhaustive computational docking procedure to determine whether the Head module can be docked unambiguously within this map. We employed the Colores search module within Situs 2.7 (*Chacon and Wriggers, 2002*) that performs a 3-stage 6D cross-correlation search of a single rigid body PDB structure within a map density at a user-defined resolution and angular sampling. Because the EM map resolution is worse than 10 Å, a Laplacian filter is applied by default to maximize the fitting contrast. We ran Colores using the map resolution value (25 Å) and threshold (0.35) reported in the EM database (using FSC 0.5 criterion) with the default angular sampling of 30°. The unnormalized correlation coefficient of the top-scoring docking solution was within the range 0.3–0.5 expected for appropriately docked Laplacian-filtered densities. The top scoring solution (unnormalized CC 0.302) was 24% higher than the next best unnormalized docking solution (unnormalized CC 0.243) and was found to be in close agreement with previous manual docking and localization studies. This agreement gave us high confidence that the Head module could be appropriate docked into the EM map at a single fixed position for the modelling study.

The EM density map was segmented into regions corresponding to the head, middle, and tail modules of Mediator by first subtracting the density associated with the docked head module and then apportioning the remaining density between middle and tail modules such that the volume of each was proportional to their molecular weight. The results were similar to the division of density proposed previously (*Tsai et al., 2014*).

## Stage 2: representation of subunits and translation of the data into spatial restraints

To maximize computational efficiency while avoiding model oversimplification, we represented the Mediator complex subunits in a multi-scale fashion. This representation employed spherical beads of varying sizes and 3D Gaussians, which coarsen domains of the complex using several resolution scales simultaneously. The spatial restraints discussed below are applied to individual resolution scales as appropriate. To improve the accuracy and precision of the model ensemble obtained by satisfaction of spatial restraints, beads and Gaussians of a given domain were arranged into either a rigid-body or a flexible string, based on the available crystallographic structures and comparative models (*Figure 2—figure supplement 1*). Within a rigid-body, all the beads and Gaussians have their relative positions constrained during configurational sampling, while in a flexible string the beads and the Gaussians are restrained by the sequence connectivity (below). The three sub-complexes determined by X-ray crystallography as well as two homology models were constrained into rigid bodies, namely: (X1) Mediator Head module ($^{1-192}$Med6, $^{23-214}$Med8, $^{4-115}$Med11, $^{182-687}$Med17, $^{2-301}$Med18, $^{2-210}$Med20, $^{1-121}$Med22) (*Robinson et al., 2012*), (X2) $^{111-206}$Med7C-$^{1-130}$Med21 (*Baumli et al., 2005*), (X3) $^{12-84}$Med7N-$^{19-110}$Med31 (*Koschubs et al., 2009*), (H1) $^{37-127}$Med4-$^{65-149}$Med9 (*Lariviere et al., 2013*), and (H2) $^{8-538}$Med16N.

Three scales were used to represent the crystallographic structures (X1, X2 and X3) and the comparative models (H1 and H2). Two of the three scales, here named fine and coarse, were set by applying two different coarse-graining criteria to the molecular volume of the atomistic structure. In both representations, the volume was decomposed into beads. For the fine scale, each bead corresponded to individual residues, and was centered at the position of the $C_\alpha$ atom. For the coarse scale, each bead represented 10-residue segments and was positioned on the center of mass of all atoms of the corresponding segment. The third scale is the Gaussian mixture model (GMM) approximation of the atomic electron density of the corresponding structure. The atomistic structures were converted into a GMM by first sampling points from the density, and then fitting the sample using the expectation-maximization algorithm implemented in scikit-learn (*Pedregosa et al., 2011*). We set the number of Gaussians to the number of residues in a component divided by 50.

We adopted a two-scale representation for the flexible strings (i.e., domains without crystallographic structures or comparative models). Each element in the string corresponded to up to 40 residues and was represented by a bead (coarse scale) and a spherical Gaussian. The fine scale representation was omitted for computation efficiency. The radius of the bead and the variance of the Gaussians were set to describe the average molecular volume and the molecular electron density of polypeptide segments, respectively. The bead and the Gaussian centers were enforced to be identical.

## Bayesian scoring function

All information gathered in Stage 1 is encoded into a Bayesian scoring function (*Rieping et al., 2005*). The likelihood function reflects the cross-linking data, while the prior depends on the sequence connectivity, excluded volume, and EM 3D restraint. Most of the remaining information (crystallographic structures of the subunits and homology models) is included in the representation, while the Mediator subunit interaction network data were used only for validating our final model.

The Bayesian approach (*Rieping et al., 2005*) estimates the probability of a model given information available about the system, including both prior knowledge and newly acquired experimental data. The model $M \equiv (X, \{\alpha_i\})$ includes the structure coordinates $X$ and additional parameters $\{\alpha_i\}$. Using Bayes' theorem, the posterior probability $p(M|D, I)$, given data $D$ and prior knowledge $I$, is $p(M|D, I) \propto p(D|M, I)p(M, I)$, where the likelihood function $p(D|M, I)$ is the probability of observing data $D$, given $I$ and $M$, and the prior is the probability of model $M$, given $I$. To define the cross-link likelihood function, one needs a forward model that predicts the data point (i.e., the presence of a cross-link between two given residues) given any model $M$, and a noise model that specifies the distribution of the deviation between the observed and predicted data points. The Bayesian scoring function is the negative logarithm of $p(D|M, I)p(M|I)$, which ranks the models identically to the posterior probability.

## Forward model

The forward model $f_n$ is computed as the probability of randomly picking two points $\tilde{r}_i$ and $\tilde{r}_j$ within the spheres centered on the C$\alpha$ atoms of the cross-linked residues, with coordinates $r_i$ and $r_j$, with unknown radii $\sigma_i$ and $\sigma_j$, such that the distance between them $\tilde{r}_{ij}$ is lower than the maximum cross-linker length $l_{XL}$; the radii $\sigma_i$ and $\sigma_j$ are proxies for the uncertainty of forming a cross-link, given structural model $X$. To reduce the number of parameters in the model, we utilized a single uncertainty parameter $\sigma$ for all residues. We imposed a maximum length of $l_{XL} = 21$ Å for the BS3 cross-link.

## Likelihood function

The BS3 cross-link dataset was used to construct a likelihood function that restrained the distances spanned by the cross-linked residues. The cross-link restraints were applied to the fine scale representation for the X-ray structures and comparative models as well as to the coarse scale of the strings.

The likelihood function for a cross-link $d_n$ is $p(d_n|X, I) = \psi \cdot (1 - f_n(X)) + f_n(X) \cdot (1 - \psi)$, where $\psi$ is the uncertainty of observing a cross-link, and is approximately equal to the fraction of inconsistent cross-links. The joint likelihood function $p(D|M, I)$ for a dataset $D = \{d_n\}$ of $N_{XL}$ independently observed cross-links is the product of likelihood functions for each data point. We promoted cross-links with high spectral redundancy, and we assigned different weight parameters to the inter- and intra-module cross-links. The goal was to lessen the impact of inter-module cross-links while benefitting from intra-module ones. Because the cross-links were collected on a sample of Mediator Holoenzyme complex, we expected the inter-module cross-links to be less accurate than the intra-module cross-links in describing the apo-mediator state. This finding is consistent with EM studies in which the presence of RNA pol II induces large motions about the hinge regions between the Mediator modules, while the intra-module topology appears unchanged (*Asturias et al., 1999*; *Davis et al., 2002*). We grouped the cross-links into two classes, including a class of inter-module cross-links (23 cross-links defined by 44 spectral matches), and a class with intra-module cross-links (271 cross-links defined by 1856 spectral matches) (*Figure 1—source data 1*).

## Priors

The model prior $p(M|I)$ is defined as a product of the priors $p(X)$ and $p(\sigma)$ on the structural coordinates $X$ and uncertainty $\sigma$, respectively. The prior $p(X)$ includes the excluded volume restraints, the sequence connectivity restraints, the EM restraint, and a weak restraint whose score depends linearly on the distance between cross-linked residues, with a slope of 0.01 Å$^{-1}$. $p(\sigma)$ is a uniform distribution over the interval [0, 100].

The excluded volume restraint was applied to the coarse scale representation for X-ray structures and comparative models as well as to the coarse scale for the flexible strings. The excluded volume of each bead was defined using the statistical relationship between the volume and the number of residues that it covered (*Shen and Sali, 2006*).

The sequence connectivity restraint imposes a harmonic upper-bound on the distance between beads that represent sequence-consecutive segments. The restraint is applied within a flexible string, or between a bead of a flexible string and a rigid-body, with a threshold distance equal to four times the sum of the radii of the two connected beads. The bead radius was calculated from the excluded volume of the corresponding residues that are represented by the bead, assuming standard protein density (*Shen and Sali, 2006*; *Fernandez-Martinez et al., 2012*).

Finally, the EM 3D restraint was imposed on the GMM representation of each domain, using the cross-correlation coefficient between GMM representations of the EM volume and model components. The weights of each GMM component were normalized to the relative mass of the component vs the mass of the module. The density of a molecule represented by a GMM is given by:

$$f(\mathbf{r}|\Theta) = \sum_{i=1}^{N} \pi_i \phi(\mathbf{r}|\boldsymbol{\mu}_i, \boldsymbol{\Sigma}_i).$$

Here $\pi_i$ are the mixing weights (normalized to 1) and $\phi(\mathbf{r}|\boldsymbol{\mu}_i, \boldsymbol{\Sigma}_i)$ is a Gaussian density function with mean $\boldsymbol{\mu}_i$ and covariance $\boldsymbol{\Sigma}_i$:

$$\phi(\mathbf{r}|\boldsymbol{\mu}_i, \boldsymbol{\Sigma}_i) = \frac{1}{(2\pi)^{3/2}|\boldsymbol{\Sigma}_i|^{1/2}} \exp\left[-\frac{1}{2}(\boldsymbol{r}-\boldsymbol{\mu}_i)^{\mathrm{T}}\boldsymbol{\Sigma}_i^{-1}(\boldsymbol{r}-\boldsymbol{\mu}_i)\right].$$

The GMM approximation for an electron density map (the data) can be calculated using the standard expectation-maximization approach using scikit-learn (*Pedregosa et al., 2011*).

The overlap function between the model (M) and the data (D) GMMs is defined by:

$$ov(\phi_M, \phi_D) = \int \phi(\mathbf{r}|\boldsymbol{\mu}_M, \Sigma_M)\phi(\mathbf{r}|\boldsymbol{\mu}_D, \Sigma_D)d\boldsymbol{r}$$

$$= \frac{1}{(2\pi)^{3/2}|\Sigma_M + \Sigma_D|^{1/2}} \exp\left[-\frac{1}{2}(\boldsymbol{\mu}_M - \boldsymbol{\mu}_D)^{\mathrm{T}}(\Sigma_M + \Sigma_D)^{-1}(\boldsymbol{\mu}_M - \boldsymbol{\mu}_D)\right].$$

This expression can be generalized as the overlap function between two GMMs:

$$ov(f_M, f_D) = \sum_{i=1}^{N_M}\sum_{j=1}^{N_D} \frac{1}{(2\pi)^{3/2}|\Sigma_{Mi} + \Sigma_{Dj}|^{1/2}} \exp\left[-\frac{1}{2}(\boldsymbol{\mu}_{Mi} - \boldsymbol{\mu}_{Dj})^{\mathrm{T}}(\Sigma_{Mi} + \Sigma_{Dj})^{-1}(\boldsymbol{\mu}_{Mi} - \boldsymbol{\mu}_{Dj})\right].$$

The cross-correlation function is (*Sfikas et al., 2005*):

$$CC(f_M, f_D) = \frac{2\int f_M(\boldsymbol{x})f_D(\boldsymbol{x})d\boldsymbol{x}}{\int (f_M^2(\boldsymbol{x}) + f_D^2(\boldsymbol{x}))d\boldsymbol{x}} = \frac{2ov(f_M, f_D)}{ov(f_M, f_D) + ov(f_D, f_D)}.$$

The negative logarithm of the cross-correlation is the EM score. We empirically found a scaling factor of 100.

## Stage 3: sampling the configurations

Structural models of the Mediator complex were computed using Replica Exchange Gibbs sampling, based on the Metropolis Monte Carlo algorithm (*Rieping et al., 2005*). The Monte Carlo moves included random translation and rotation of rigid bodies (up to 2 Å and 0.04 radians, respectively), and random translation of individual beads in the flexible segments (up to 3 Å). 64 replicas were used,

with temperatures ranging between 1.0 and 2.5. 10 independent sampling calculations were performed, each one starting with a random initial configuration. A model was saved every 10 Gibbs sampling steps, each consisting of a cycle of Monte Carlo steps that moved every rigid body and flexible bead once. The sampling produced a total of 165,523 models in 20 independent runs. Additionally, we performed 34 sampling runs by randomly removing (i.e., 'jackknifing') 10% of the cross-links, thereby producing 198,498 models. The entire sampling procedure took approximately 2 weeks on a cluster of 1280 computational cores.

## Stage 4: analysis and assessment of the ensemble

First, the exhaustiveness of configurational sampling was assessed by comparing two subsets of the whole ensemble of sampled models. The 500 best scoring models from runs 1–10 (first half) and the 500 best scoring models from runs 11–20 (second half) were converted into localization density maps (definition below) (*Figure 2—figure supplement 4*). The localization density map of a subunit was contoured at a density of 0.15. Importantly, the two localization density maps were similar to each other, and to the density map obtained considering the whole ensemble of models, demonstrating that the Monte Carlo algorithm likely sampled all solutions that satisfy the input restraints. The final localization density maps of the Mediator subunits and the whole complex were computed from the complete ensemble solutions (i.e., the 500 best scoring models considering all 20 sampling runs).

Second, the ensemble of solutions was assessed in terms of how well they satisfy the data from which they were computed, including the cross-links as well as the excluded volume, sequence connectivity, and the 3D EM restraints. We validated the ensemble of solutions by comparing it with the ensemble of solutions obtained jackknifing the cross-link dataset. The obtained localization maps are similar to the one computed without jackknifing the dataset (*Figure 2—figure supplement 4*), suggesting that the cross-link data are accurate and the models are not a result of overfitting. The 3D shape implied by the EM restraint was satisfied by the ensemble. The excluded volume and sequence connectivity restraints were also satisfied.

Third, the solutions (500 best scoring models) were grouped by RMSD k-means clustering, based on the position of the beads representing all subunits of the Mediator complex. Owing to the mixed resolution of the representation, consisting of coarse-grained beads comprising 20–40 residues as well as individual residues, the RMSD was computed by weighting the average squared displacement by the size of the beads:

$$RMSD_{ij} = \left( \frac{\Sigma_b n_b \left( \vec{x}_{b,i} - \vec{x}_{b,j} \right)^2}{\Sigma_b n_b} \right)^{1/2},$$

where $\vec{x}_{b,i}$ and $\vec{x}_{b,j}$ are the coordinates of bead $b$ in models $i$ and $j$, respectively, and $n_b$ is the number of residues represented by the bead.

The precision of each cluster was calculated as the average RMSD between the individual solutions and the cluster-center solution, defined as the solution with the minimal sum of the RMSD's to the other solutions in the cluster (*Figure 2—figure supplement 4*). The localization density maps of the clusters were computed as described above.

Finally, we performed the following analysis:

## Localization density map (*Figures 2, 4–7*, *Figure 2—figure supplements 3–4*)

For a given cluster of solutions, we computed the probability of finding a given protein at any point in space (i.e., the localization density map). All localization density maps of proteins and domains are represented by an isosurface, corresponding to the threshold of 0.15 unless otherwise stated.

## Contact-map (*Figures 4, 5*)

The proximities of any two residues in each cluster were measured by their relative 'contact frequency', which is defined by how often the two residues contact each other in the cluster; a pair of residues are in contact when the distance between the surface of the fine resolution beads is less than 10 Å.

## Domain–domain interaction map (*Figure 3*)

Two domains are in contact when the surface of any bead of one domain is within 10 Å from the surface of any of the beads of the other domain. Long sequences are divided into domains of 200 residues. The domain–domain interaction map displays the frequency of contacts between every pair of domains for a given cluster of solutions.

## Precision (*Figure 2—figure supplement 3F*, *Figure 2—figure supplement 4C–E*, *Figure 2—figure supplement 5*)

The precision of a domain (or a protein, or the whole complex) is calculated as the average RMSD (computed on the beads of the corresponding domain, or protein, or whole complex) between the cluster center and all other solutions in the cluster.

## Average distance between clusters (*Figure 2—figure supplement 4C–E*)

The average distance between two clusters is calculated as the average RMSD, computed on the whole complex, between every pair of solutions taken from the two clusters.

## RMSF (*Figure 2—figure supplement 5A*)

RMSF of a given residue is the standard deviation of distances between the position of the residue in each solution of the cluster and its position in the cluster center. The position of the residue is the center of the bead that was used to represent it.

Molecular visualizations were prepared using the UCSF chimera package (*Pettersen et al., 2004*).

# Selection of the cluster of solutions

In the IMP modeling procedure we perform a large number of independent modeling runs, each starting from a random configuration and undergoing thousands of rounds of small random model perturbations, in order to achieve exhaustive sampling of conformational space. As we are interested only in models that best satisfy the restraints, we select a small fraction of the models representing the very best scoring solutions (*Figure 2—figure supplement 3A*). These models then undergo pairwise-RMSD clustering to separate out divergent structural states (*Figure 2—figure supplement 3B*). In our study, most of the best scoring models (92%, n = 500) gave a unique subunit arrangement for 18 of the 21 Mediator subunits. The divergent clusters differed only in the relative locations of Med5, Med15 and Med16 within the Tail module (*Figure 2—figure supplement 3C*). We compared these three clusters in terms of crosslink violation statistics (*Figure 2—figure supplement 3E*) and consistency with prior EM localization experiments (*Figure 2—figure supplement 3H*). A single cluster showed both consistency with earlier localizations and significantly better violation statistics and was therefore deemed to be the Mediator architecture with the highest confidence.

# Molecular architecture of the RNA Pol II complex revealed by integrative modeling

Modeling studies of RNA pol II employed cross-link data from two sources: 156 cross-links from previous experiments conducted with the free pol II fraction isolated during our holoenzyme SEC purification (*Trnka et al., 2014*), and 108 cross-links from a previous study (*Chen et al., 2010*). These sources had 63 cross-links in common. Hence, the current study used 201 unique cross-links. The atomic structures for the twelve-subunit yeast pol II have been previously determined by X-ray crystallography (PDB accession ID: 1WCM, [*Armache et al., 2005*]). The crystallographic structure covers 89% of the residues in the pol II complex. A 20.9 Å resolution density map from single particle EM reconstruction of the pol II complex was also considered (EMDB accession id: 1883, [*Czeko et al., 2011*]). The 12 pol II subunits were represented as 15 domains, where Rpb1 was decomposed into 3 domains (residues: 1–1140, 1141–1274, 1275–1733), Rpb2 was decomposed into 2 domains (residues 1–1102, 1103–1224), and the remaining subunits were represented as single domains. Regions without a crystal structure were modeled as beads containing up to 5 amino acid residues. We followed the same 4-stage procedure as described for the Mediator complex above.

## Acknowledgements

We thank JH Morris, CC Huang, EC Meng and other members of the Resource for Biocomputing, Visualization, and Informatics at UCSF for support with Cytoscape and Chimera. We acknowledge B Webb for his help with IMP, J Baker-LePain for his help with the cluster at UCSF, as well as D Saltzberg and P Cimermancic for discussions. This work was supported by the US National Institutes of Health (NIH) grants R01 AI21144 (to RK), P41 GM109824 and R01 GM083960 (to AS), and P41 GM103481 (to AB). We also acknowledge support from Human Frontier Science Program long-term fellowship LT00160 (to PJR) and the US National Science Foundation through partnership in the BioXFEL Science Technology Center supported by grant NSF-1231306 (to RK). Yeast fermentation was performed using an instrument purchased using funds from the NIH S10 shared instrumentation grant S10RR028096.

## Additional information

### Funding

| Funder | Grant reference | Author |
| --- | --- | --- |
| National Institutes of Health (NIH) | GM 109824 | Andrej Sali |
| National Institutes of Health (NIH) | GM 083960 | Andrej Sali |
| National Institutes of Health (NIH) | GM 103481 | Alma L Burlingame |
| National Science Foundation (NSF) | 1231306 | Roger D Kornberg |
| National Institutes of Health (NIH) | S10RR028096 | Roger D Kornberg |
| Human Frontier Science Program (HFSP) | LT00160 | Philip J Robinson |
| National Institutes of Health (NIH) | AI21144 | Roger D Kornberg |

The funders had no role in study design, data collection and interpretation, or the decision to submit the work for publication.

### Author contributions

PJR, MJT, RP, Conception and design, Acquisition of data, Analysis and interpretation of data, Drafting or revising the article; CHG, Analysis and interpretation of data; DAB, Conception and design, Acquisition of data, Analysis and interpretation of data; RD, Performed Holoenzyme reconstitution experiments that were critical to determine that the native form of the Holoenzyme is more stable than the reconstituted counterpart; ALB, AS, RDK, Conception and design, Analysis and interpretation of data, Drafting or revising the article

## Additional files

### Supplementary file

• Supplementary file 1. Full list of Mediator–Mediator cross-linked peptides discovered by mass spectrometry. Separate Microsoft Excel File.

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
