## [Decision Letter]

Thank you for submitting your work entitled “Molecular Architecture of the Yeast Mediator Complex” for peer review at *eLife*. Your submission has been favorably evaluated by James Manley (Senior Editor) and four reviewers, one of whom, Irwin Davidson, is a member of our Board of Reviewing Editors. The reviewer Bruno Klaholz has also agreed to share his identity.

The reviewers have discussed the reviews with one another and the Reviewing editor has drafted this decision to help you prepare a revised submission.

The manuscript by Robinson et al. addresses the structural organization of the mediator complex. This multi-subunit complex acts in concert with the RNA polymerase to regulate transcription, notably initiation, in eukaryotes. Current direct structural data of the holoenzyme complex are limited to a low-resolution cryo-EM map and crystal structures of some subunits or domains. By gathering additional indirect structural information through chemical cross-linking and mass spectrometry (MS) the authors derive a molecular model through integrative modelling in which the subunits are placed taking into account previously available data, in particular the existing cryo-EM map at 25 Å resolution of the yeast mediator holoenzyme complex. To validate the modelling procedure, the authors used the RNA polymerase part of the complex for which the crystal structure is known. Taken together, the new MS data (about 200 unique cross-links) and the impressive integrative modelling offer a useful molecular model of the mediator complex providing many new insights into the complex organisation and inter-subunit interactions. Some caution though should be taken considering that this is mostly based on indirect rather than direct structural evidence, and this should be clarified to the general reader. It should also be stressed that the major steric constraints used to compute the model come from the envelope of the Mediator that was obtained previously by Tsai et al. (Cell 2014) and the X-ray structure of Mediator subunits (several papers).

Essential revisions:

The following points should be seriously addressed by the authors.

1) Can the authors comment on the confidence that can be placed on the architecture of the tail module, which is a flexible structure and for which no crystallographic data is available. The predicted structure resolution of the head and middle modules is rather low, compared to those that were published by Tsai et al. and [45].

2) Plaschka et al. reported in 2015 a structure of a complex of Pol II, Mediator head and middle modules, TBP, TFIIB, TFIIF and DNA at 9.7 Å. The authors compare the cross-links they have obtained with their Pol II-Mediator complex with those that were published by Plashka et al. They show that 17 Mediator to Pol II cross-links are largely incompatible with the position of Mediator modules found in the core initiation complex. Given that the authors have the necessary data they should compute a model for the Mediator-Pol II complex and present this data in comparison with that of Plaschka. It would make sense to have the RNA polymerase included in the complex description because (i) the complex was purified as a holoenzyme complex including both mediator and RNA polymerase, and (ii) the RNA polymerase has been used as an internal control.

3) The authors should describe the precision of the multi-parameter refinement, in particular how far alternative solutions are away from the proposed model. To have some measure of confidence they should list the subunits for which the solutions are non-unique (if any), and possibly create a 3D confidence map. While the RNA polymerase structure was used for internal validation the high RMSD also illustrates the relatively low precision of the method. Furthermore, an average RMSD of 14 Å, if applicable to the entire complex implies a much lower precision than required to annotate general secondary structure elements. This contrasts with the potentially misleading statement that “residue-level resolution” was reached, which is only true for residues with direct cross-link information. It would therefore be good to clarify these aspects to the reader and somehow down tune the general statement (notably in the Abstract, in the subsection “A Single, Best-Scoring Subunit Architecture”, and in the Discussion).

4) The derived molecular model and the 3D annotation of the subunits and interfaces within the complex should be made available to other scientists for future studies by depositing the coordinates in the protein data bank. Access to the indicated website with all of the data and analysis scripts should be made available publically if the manuscript is accepted.

5) In the subsection “Validation of Holoenzyme Cross-link Dataset”, specify the resolution (0.5 and 0.143 criteria) of the cryo-EM map used for modeling, and the EMD ID.

6) In the third paragraph of the subsection “Validation of Holoenzyme Cross-link Dataset”, regarding docking solutions, does the observation that the next-best solution is 40% worse apply only to the analysis of the head module? How unique are the solutions for the individual mediator subunits (see comment 3 above)?

7) In the last paragraph of the subsection “Integrative modeling of Mediator complex”, please clarify whether such conformational changes can really be assessed considering the relatively low precision of the method.

[Editors' note: further revisions were requested prior to acceptance, as described below.]

Thank you for resubmitting your work entitled “Molecular Architecture of the Yeast Mediator Complex” for further consideration at *eLife*. Your revised article has been favorably evaluated by James Manley (Senior Editor), a Reviewing Editor, and two reviewers. The manuscript has been improved but there are some remaining issues that need to be addressed before acceptance, as outlined below. The revisions required concern additions to the current text.

Since the complex was purified as a holoenzyme complex including both mediator and RNA polymerase, it should be mentioned at the beginning of the manuscript why the data describe the mediator rather than the holoenzyme complex.

Please include some sections from the answer to referees to the Methods section.

The following information should be included:

a) “The average precision for the localization of Tail module beads is 33.5 Å compared to 24.4 Å for the Middle module (see Figure 2—figure supplement 4 and Figure 2—figure supplement 5). However, such average precision values are entirely sufficient to confidently position the Tail module components within the large Tail EM density (208 x 123 x 98 Å), especially as certain key architectural features with high crosslink density and consequently high local precision values (such as the Middle-Tail junction), themselves provide powerful positional constraints that indirectly effect the localization of otherwise less constrained residues.”

b) “Plaschka map represents only ∼50% of the full Mediator complex mass as they do not include either the 480KDa Tail module or the Middle module protein Med1.” (To be reformulated).

c) Clarify the answer to point 6 on the IMP procedure and the selection of solutions.

---

## [Author Response]

*1) Can the authors comment on the confidence that can be placed on the architecture of the tail module, which is a flexible structure and for which no crystallographic data is available. The predicted structure resolution of the head and middle modules is rather low, compared to those that were published by Tsai et al. and*
[45]*.*

A comparison of our precision values to the resolution of EM maps of Tsai et al. and Plaschka et al. is inappropriate. The precision values reported for Mediator components in our model represent average fluctuations across individual atoms or beads localized in 3D, whereas the resolution criterion for the EM maps represents the uncertainty of the overall electron density, with no relationship to the placement of individual atoms, beads, or subunits.

As noted in the manuscript, the main source of variation among the top scoring models lay in the organization of the Tail module (which was modelled at the 40 residue/bead resolution level). We had to distinguish among three alternative models (Figure 2—figure supplement 3), which we could do on the basis of average score, representing the satisfaction of all modelling constraints, and on the basis of the cross-link violation statistics, which were significantly better (student t-test) in the case of the top scoring solution (see Figure 2—figure supplement 3). The top scoring solution was also consistent with the subunit localizations suggested by [58] (Figure 2—figure supplement 3). Taken together, these results give high confidence in our proposed Tail module architecture. We also performed additional quality tests to assess the model sampling convergence and the robustness of the cross-link dataset (Figure 2—figure supplement 4). First, since the arrangements of the subunits were identical when we took the whole ensemble of models, and two randomly chosen subsamples that comprises 50% of the models, ruled out any limited-sampling artefact. Second, the same subunit arrangement was reproduced by cross-link dataset jackknifing (i.e., randomly removing 10% of the crosslinks), demonstrating the robustness of the crosslink dataset, which is critical in determining the architecture of the whole complex.

As noted by the reviewers, the lack of atomic models for the very large Tail module proteins combined with occasional regions of sparse crosslink coverage (e.g. C-termini of Med2 and Med3) does lead to a modest increase in positional ambiguity compared to other regions of the architectural model. Accordingly, the average precision for the localization of Tail module beads is 33.5 Å compared to 24.4 Å for the Middle module (see Figure 2—figure supplement 4 and Figure 2—figure supplement 5.). However, such average precision values are entirely sufficient to confidently position the Tail module components within the large Tail EM density (208 x 123 x 98 Å), especially as certain key architectural features with high crosslink density and consequently high local precision values (such as the Middle-Tail junction), themselves provide powerful positional constraints that indirectly effect the localization of otherwise less constrained residues. The residue-level and overall precision of the tail module subunits are reported in Figure 2—figure supplement 5.

To clarify these points in the manuscript we have added text to the Results section (subsection “A Single, Best-Scoring Subunit Architecture”).

2) Plaschka et al. reported in 2015 a structure of a complex of Pol II, Mediator head and middle modules, TBP, TFIIB, TFIIF and DNA at 9.7 Å. The authors compare the cross-links they have obtained with their Pol II-Mediator complex with those that were published by Plashka et al. They show that 17 Mediator to Pol II cross-links are largely incompatible with the position of Mediator modules found in the core initiation complex. Given that the authors have the necessary data they should compute a model for the Mediator-Pol II complex and present this data in comparison with that of Plaschka. It would make sense to have the RNA polymerase included in the complex description because (i) the complex was purified as a holoenzyme complex including both mediator and RNA polymerase, and (ii) the RNA polymerase has been used as an internal control.

What the reviewers have requested is impossible. First, contrary to the reviewers’ statement, there is no cryo-EM map of the Mediator-Pol II complex available for modelling by our procedure. We employed a map of Mediator alone, the only map suitable for our study, at the time or at present. Secondly, our Mediator model can be docked into EM map of Plaschka et al., explaining the 12 Mediator-Pol II cross-links of Plaschka et al., but our 17 Mediator – Pol II cross-links are incompatible with their structure, demonstrating a major rearrangement between the Mediator – Pol II complex in our work and the initiation complex of Plaschka et al. (see Figure 7). Thirdly, the Plaschka map represents only ∼50% of the full Mediator complex mass as they do not include either the 480KDa Tail module or the Middle module protein Med1.

It must also be emphasized that 90% of our cross-links are between protein subunits in the same module, whereas only 6% are between proteins in different modules, and only 4% are between Mediator subunits and Pol II. Our data therefore provide a strong basis for modeling the organization or subunits within modules, but not for modeling the Mediator – Pol II interaction.

3) The authors should describe the precision of the multi-parameter refinement, in particular how far alternative solutions are away from the proposed model. To have some measure of confidence they should list the subunits for which the solutions are non-unique (if any), and possibly create a 3D confidence map. While the RNA polymerase structure was used for internal validation the high RMSD also illustrates the relatively low precision of the method. Furthermore, an average RMSD of 14 Å, if applicable to the entire complex implies a much lower precision than required to annotate general secondary structure elements. This contrasts with the potentially misleading statement that “residue-level resolution” was reached, which is only true for residues with direct cross-link information. It would therefore be good to clarify these aspects to the reader and somehow down tune the general statement (notably in the Abstract, in the subsection “A Single, Best-Scoring Subunit Architecture”, and in the Discussion).

The reviewers’ concern calls for clarification, both in response and by revision of the manuscript. The precision of our model reflects not a limitation of our method but rather of the input data. The model reflects the sparseness and uncertainty of the input data, without over-interpretation. In the example of Pol II, we derived a model from 201 cross-links and a 21 Å resolution EM map. Because the input data was sparse and noisy, an ensemble of solutions was obtained. The precision of the model (defined above) of 14 Å was nevertheless sufficient to unambiguously determine the positions of the Pol II subunits in the map. The analysis of Pol II was performed in this way to enable comparison with our study of Mediator, which employed a 25 Å resolution EM map. The precision of placement of subunits in the best model is addressed in Figure 2—figure supplement 5. There we represent the root mean square fluctuation (RMSF) of each non-Head module subunit across the entire length of the subunit, both as a graphical plot and as a 3-D heat map, in which gradients of color are used to display regions of higher and lower localization precision. The term “residue-level resolution” indeed applies only to cross-linked residues, and we have modified the text to clarify this point in the Abstract, in the subsection “A Single, Best-Scoring Subunit Architecture”, and in the Discussion.

4) The derived molecular model and the 3D annotation of the subunits and interfaces within the complex should be made available to other scientists for future studies by depositing the coordinates in the protein data bank. Access to the indicated website with all of the data and analysis scripts should be made available publically if the manuscript is accepted.

The modeling data and scripts will be made publicly available as indicated, and as we have done in the past https://integrativemodeling.org/systems/. We believe that a localization density representation, accessible through an MRC format (most similar to a fully segmented EM map), will be most useful. Additionally, we will provide access to the Rich Molecular Format (RMF, http://integrativemodeling.org/rmf/nightly/doc/format.html) files of the models (which can be open using Chimera) and the corresponding PDB files.

5) In the subsection “Validation of Holoenzyme Cross-link Dataset”, specify the resolution (0.5 and 0.143 criteria) of the cryo-EM map used for modeling, and the EMD ID.

EMDB details were added for Pol II EM map (in the subsection “Validation of Holoenzyme Cross-link Dataset”) and for Mediator EM map (“Exhaustive docking of Head Module within EM density”).

6) In the third paragraph of the subsection “Validation of Holoenzyme Cross-link Dataset”, regarding docking solutions, does the observation that the next-best solution is 40% worse apply only to the analysis of the head module? How unique are the solutions for the individual mediator subunits (see comment 3 above)?

The Situs docking study was performed to confirm computationally that there was one Head module docking solution into the cryo-EM map that stood apart from all alternatives as an “unbiased” correct solution. Differences of 40% in Situs between the best and next-best solutions are considered highly significant.

In the IMP modeling procedure we perform a large number of independent modelling runs, each starting from a random starting configuration and undergoing thousands of rounds of small random model perturbations, in order to achieve full sampling of conformational space. After each small perturbation, the model is scored for how well it satisfies the input modeling restraints. As we are interested only in models that best satisfy the restraints, we select the very small fraction of the models representing the very best scoring solutions. These models then undergo RMSD pairwise clustering to separate out divergent structural states. In our study, most of the models in the best scoring group (92%, n=500) gave a single unique subunit arrangement for 18 of the 21 Mediator subunits. As stated above and in the manuscript, the divergent clusters differed only in the Tail module, which were subsequently resolved with high confidence.

To highlight the differentiation of the best model clusters from other configurations, we have plotted the score landscape for the entire population of scored models and shown the cutoff threshold for the best 500 models chosen for cluster analysis. This graphical representation of score distribution has been appended to Figure 2—figure supplement 3.

7) In the last paragraph of the subsection “Integrative modeling of Mediator complex”, please clarify whether such conformational changes can really be assessed considering the relatively low precision of the method.

This point relates to crosslinking violation statistics rather than a reflection of the precision of the modeling procedure. As stated in the aforementioned paragraph, an analysis of the satisfaction of crosslinks in the best scoring cluster shows that whereas intra-modular Mediator crosslinks (residue-residue distances within modules) are 95% satisfied in this cluster, inter-modular crosslinks (residue-residue distances between modules) are only 10% satisfied. This statistic alone is strongly suggestive that the configuration of modules in the free Mediator state (Asturias cryo-EM map) is not consistent with that found in the Holoenzyme state.

[Editors' note: further revisions were requested prior to acceptance, as described below.]

Since the complex was purified as a holoenzyme complex including both mediator and RNA polymerase, it should be mentioned at the beginning of the manuscript why the data describe the mediator rather than the holoenzyme complex.

This issue has been addressed by the addition of a descriptive paragraph in the subsection “Integrative modeling of Mediator complex”, immediately preceding our description of the modeling pipeline.

Please include some sections from the answer to referees to the Methods section. The following information should be included:

*a) “The average precision for the localization of Tail module beads is 33.5 Å compared to 24.4 Å for the Middle module (see*
Figure 2—figure supplement 4 and Figure 2—figure supplement 5*). However, such average precision values are entirely sufficient to confidently position the Tail module components within the large Tail EM density (208 x 123 x 98 Å), especially as certain key architectural features with high crosslink density and consequently high local precision values (such as the Middle-Tail junction), themselves provide powerful positional constraints that indirectly effect the localization of otherwise less constrained residues.”*

A slightly modified form of the excerpt above was added to the relevant section of main text (subsection “A Single, Best-Scoring Subunit Architecture”).

b) “Plaschka map represents only ∼50% of the full Mediator complex mass as they do not include either the 480KDa Tail module or the Middle module protein Med1.” (To be reformulated).

A slightly modified form of the excerpt above was added to the relevant section of main text (“Holoenzyme Configuration”).

c) Clarify the answer to point 6 on the IMP procedure and the selection of solutions.

A new section entitled ‘Selection of the cluster of solutions’ has been added to the Materials and methods section. The new section is based largely on our answer to point 6 in the previous response to the reviewers’ comments.